# Biomass Price Prediction Based on the Example of Poland

Aleksandra Górna [1], Marek Wieruszewski [2,*], Alicja Szabelska-Beręsewicz [3], Zygmunt Stanula [1] and Krzysztof Adamowicz [1]

[1] Department of Forestry Economics and Technology, Faculty of Forestry and Wood Technology, Poznan University of Life Sciences, Wojska Polskiego 28, 60-637 Poznan, Poland

[2] Department of Mechanical Wood Technology, Faculty of Forestry and Wood Technology, Poznan University of Life Sciences, Wojska Polskiego 28, 60-637 Poznan, Poland

[3] Department of Mathematical and Statistical Methods, Poznan University of Life Sciences, Wojska Polskiego 28, 60-637 Poznan, Poland

* Correspondence: marek.wieruszewski@up.poznan.pl

**Abstract:** The aim of the study was to test the applicability of forecasting in the analysis of the variability of prices and supply of wood in Poland. It relies on the autoregressive integrated model (ARIMA) that takes into account the level of cyclic, seasonal, and irregular fluctuations and the long-term trend as tools for the assessment of the predictions of the prices of selected medium-sized wood assortments. Elements of the time series were determined taking into account the cyclical character of the quarterly distribution. The data included quarterly information about the supply (amount) and prices (value) of wood sold by state forests in the years 2018–2022. The analysis was conducted for the most popular assortments: logging slash (M2, M2ZE), firewood S4, and medium-sized wood S2AP. In the period studied (years 2018–2022), the average rate of price variation was widely scattered. The average rate of price variation for the M2ZE assortment amounted to 7%. The average rate for M2 assortment was 1%, while the medium-sized S2AP assortment displayed the greatest variation of 99%. This means that between 2018 and the present, the price increased by nearly 100%. No major fluctuations were observed for the S4 assortment and its average rate of variation amounted to 0%. The analysis found seasonal variation was observed only for S4 firewood, the price of which went up each year in October, November, and December. For this reason, the forecast was made with the seasonal autoregressive integrated moving average (SARIMA) version of the model. It is difficult to forecast the price of wood due to variations in the market and the impact of global factors related to fluctuations in supply.

**Keywords:** wood market; wood prices; forecasting; seasonality of supply; slash; pulpwood; ARIMA; SARIMA

## 1. Introduction

In the process of forest resource management, the amount of wood harvested and its assortment structure should be optimised to adapt to market needs, depending on the changing economic conditions and considering changes in the European economy that take climate change into account in long-term strategies [1]. Empirical studies on the wood industry tend to focus on developing long-term supply scenarios, while globalisation has given rise to links between neighbouring wood markets, which implies that the domestic market responds to macroeconomic conditions [2]. Therefore, the significance of short and long-term economic forecasts [3,4] for the wood market is increasing. The analysis of supply on wood markets involves uncertainty related to climate change, the dying out of forest stands, and disaster risk. Other factors at play include technological progress in wood harvesting, as well as political and economic programmes resulting in legal restrictions [5,6]. Furthermore, regional aspects play an increasingly important role, and wood harvesting may be restricted due to the growing awareness of the social functions of forests [7–9].

Wood availability is a significant problem at the stage of designing development strategies for forest and wood areas when climate change and the role of forests in the process of carbon accumulation are taken into account [10,11]. Market prices for the available wood assortment structure change under the influence of multiple factors [12–14]. The prices depend, i.e., on demand from the wood industry, which, in turn, is influenced by the domestic and global economic climate [15,16]. In Poland, the supply of the resource is influenced by the available structure of assortments offered by State Forests—National Forest Holding (PGL LP). PGL LP is the dominant entity engaged in the supply of wood. It offers wood according to the accepted terms of sale set out in the legislation that determines detailed sales procedures and minimum prices. PGL LP is a non-corporate entity responsible for managing state forests in Poland. It has its own independent budget, and 90% of its revenue comes from the sale of wood. Coniferous forests account for more than half of the total forest area in Poland, while pine forests prevail at 60.2% of that area [17]. The Scots pine coexists with other tree species of economic significance, such as the European spruce and European silver fir (6.1% and 6.5%, respectively). Deciduous trees include the common oak and Cornish oak (7.7%), silver birch (7.3%), common beech (5.9%), and black alder (5.7%) [18]. Furthermore, wood harvesting is limited by numerous other forms of nature conservation, such as reserves and protected zones for animal and plant species [19]. The activity of PGL LP is subject to numerous legal regulations, especially with respect to the development and implementation of 10-year forest planning and management plans concerning the harvesting of wood as part of cutting and silvicultural measures. In Poland, wood is sold on the free market; however, PGL LP applies detailed procedures for entering into a purchase determined by the Director General of LP. Each year, the wood is divided into assortment and species groups and is offered in several procedures: online sale via the Wood and Timber Portal, "e-wood", or through special auctions of valuable wood assortments, commercial negotiations, and on the basis of a retail price list. The Director General also makes decisions concerning initial prices and conversion coefficients for specific assortment and species groups. Detailed rules for the sale of wood are regulated by decisions of the Director General of the State Forests [20] that regulate the rules for signing contracts with enterprises as part of the so-called "purchase history" and free access to purchases at higher price levels [21,22]. In recent years, the share of small wood M2 and medium-sized wood S2A assortments (25%) in the total amount of wood sold by PGL LP has been on the increase (M2ZE-branch and chipped wood destined for woodchips for energy purposes with a minimum diameter of at least 5 cm without bark (7 cm in bark), the length or quality of which does not allow industrial use; M2 small-sized firewood up to 7 cm in diameter with a length of 0.5 m to 6.0 m; S4 medium-sized firewood up of 5 cm with a length of 0.5 m to 6.0 m) [20]. These assortments are most often used to produce heat energy in individual households as well as in biomass power plants. Due to the need for new sources of energy, where wood and wood residues are recognised as RES, the demand for these goods has definitely increased. The disaster situation caused by the bark beetle had an impact on the increased supply of biomass assortments; however, this was local in nature. Increased awareness of the crisis may have contributed to an increased demand for energy wood.

According to the regulations, wood prices are not the primary criterion regulating the amount of wood available on the market [23]; however, they are significant for optimising forest management planning [24,25]. Adjusting prices to the level of supply can be considered for optimising the benefits of price fluctuations [23,26]. Information on changes in wood prices is a source of knowledge used in the planning of the activities of timber companies and in the strategic planning of forest management. Therefore, studies explaining the mechanisms of shaping the wood market through changes in wood prices over time are the subject of numerous publications in Poland and in other countries [27–46]. Time series are used to consider the price performance of the wood market [47,48]. The application of appropriate predictive methods makes it possible to forecast changes in

wood prices [49–54]. Currently, most studies focus on analysing changes or trends in time series and on their role in the forecasting of market prices [55].

The forecasting of wood prices and supply has a long history in forest economics, with most studies being based on models of supply and demand in various geographical regions [42]. Flexibility in planning wood prices is especially important in the face of random events, such as windfalls or sudden economic changes. Meanwhile, in times of intensive economic growth, it is important to ensure a sufficient and stable supply of wood. The supply of wood influencing prices of the raw material is characterised by deviation from the planned levels. Depending on the cause, these changes can be divided into: irregular fluctuations (e.g., due to large-scale disasters), long-term trends [3], periodic fluctuations linked to the economic cycle, and seasonal fluctuations. Explaining the mechanisms behind wood price fluctuations is important for the revenues of forestry operators worldwide and for their customers (stability of supply), as well as for enterprises providing services related to wood harvesting and forest care. An element of inflationary factor in the price of the raw material is included. The authors used data in which this factor affects the proposed price. The effect of inflation on the price of timber can be inferred from the incurred cost of timber, which has increased under the influence of political and economic factors (e.g., the price of fuel, the cost of machinery, etc.).

The aim of this study was to apply temporal predictive models to identify fluctuations in irregular, cyclic, and long-term trends in the prices of small- and medium-size wood offered for sale by PGL LP in Poland.

## 2. Materials and Methods

The study involved the use of a predictive methodology that employs the autoregressive integrated moving average (ARIMA) model among its provisions. The topic of forecasting the price of wood raw material in Poland is an evolving one. The choice of ARIMA methodology is covered by numerous literature items, where the method has been used in many other fields, including with a strongly changing external area. Using the ARIMA ($p$, $d$, $q$) model requires the estimation of $p$, $d$, and $q$ values. The meaning of these parameters is as follows:

$p$ is the autocorrelation parameter
$d$ stands for the degree of integration of a series
$q$ is the parameter of the moving average

The general ARIMA model can be presented as follows in Equation (1):

$$\varphi(B)\nabla^d z_t = \theta(B)a_t \ \ or \ \ z_t = \sum_{i=0}^{p} \varnothing_i z_{t-1} + \alpha_t - \sum_{k=1}^{q} \theta_i a_{t-k} \tag{1}$$

$\alpha_t -$ random disturbance; $z_{t-1} -$ random disturbance at the moment $t-1$.

where Equation (2):

$$\varphi(B) = 1 - \varphi_1 B - \varphi_2 B^2 - \ldots - \varphi_p B^p \tag{2}$$

is a function related to the non-seasonal autoregressive parameter $p$, and where Equation (3):

$$\theta(B) = 1 - \theta_1 B - \theta_2 B^2 - \ldots - \theta_q B^q \tag{3}$$

is the function associated with the associated non-seasonal moving average parameter $q$ and the parameter $B$ as the backward shift operator, which we can write as Equation (4):

$$B^p Z_t = Z_{t-p} \tag{4}$$

$\varphi_1, \varphi_2, \ \varphi_p \ \ldots oraz \ \theta_1, \theta_2, \ \theta_q$ are unknown coefficients estimated on the basis of example data using approximate probability.

$\nabla^d = (1 - B)^d$, $d$ is the backward shift operator of the p-th order, defined as Equation (5):

$$\nabla Z_t = Z_t - Z_{t-1} \ \text{with} \ \nabla^d = \nabla \nabla^{d-1} \tag{5}$$

The SARIMA $(p, d, q)$ $(P, D, Q)$ model is an ARIMA model that takes into account the seasonality component (P—order of seasonal lags of the AR type, Q—order of seasonal lags of the MA type, D—differentiation of the seasonal component).

The mean absolute error (MAE) is calculated by:

$$\Delta x = |x - x_0| \tag{6}$$

$x$—real value of price; $x_0$—forecast value of prie; The mean absolute percentage error (MAPE) is calculated by:

$$\delta = \frac{|x - x_0|}{x} \times 100\% \tag{7}$$

The root mean squared error (*RMSE*) is calculated by:

$$RMSE = \sqrt{\frac{\sum |x - x_0|}{N}} \tag{8}$$

$N$—number of observation.

Checking data integration is recognised as the first stage of the calculation. This means determining whether the data are stationary or can be converted to stationarity. When the data to be processed are stationary, we adopt the parameter $d = 0$; when the data need to be converted, parameter $d > 0$. The Kwiatkowski–Phillips–Schmidt–Shin (KPSS) [56] test was used to determine data stationarity. The test verifies the hypothesis whether the process is stationary, in other words, whether it is expedient to take differences into account. As we begin to consider stationarity, we first subject raw data to the test. Data differencing processes are carried out until the data are stationary following the KPSS test.

To determine the other parameters $p$ and $q$, the autocorrelation function (ACF) and partial autocorrelation function (PACF) are useful. Based on the ACF and PACF graphs, we can proceed to identify the autocorrelation parameter and the moving average. By comparing the graphs, we determine the growth levels of the data in the adopted periods and determine the likely variant of the ARIMA model.

In the study, variants of $p$, $d$, and $q$ parameters for each set of data for specific assortments were determined separately. The calculations were made with the "R" software, version 4.0.3 released in 2022 (GNU General Public License as published by the Free Software Foundation).

Data in the study were divided into individual assortments, such as residual wood M2ZE, firewood slash M2, medium-sized firewood S4 with the permissible level of soft rot set at 50%, minimum top diameter of 5 cm without bark, and general-purpose cordwood S2AP with the permissible level of rot at up to 50%, that meet the requirements complying with the definition of energy wood included in the directive 2018/2001 of the European Parliament and of the Council of 11 December 2018 as a source of renewable energy. The data include quarterly data for the period 2018—second quarter of 2022, which provides exactly 18 observations for the forecast. The forecast was made for the three following quarters, more specifically, for Q3 of 2022, Q4 of 2022, and Q1 of 2023 (Figure 1).

The authors used data in which this factor affects the proposed price, which means using nominal prices. The study aims to forecast real prices. The additional inflation factor will not reflect actual prices.

The sensitivity analysis was performed in comparison with the third quarter of 2022, as this period can be considered closed, and PGL LP reported the data.

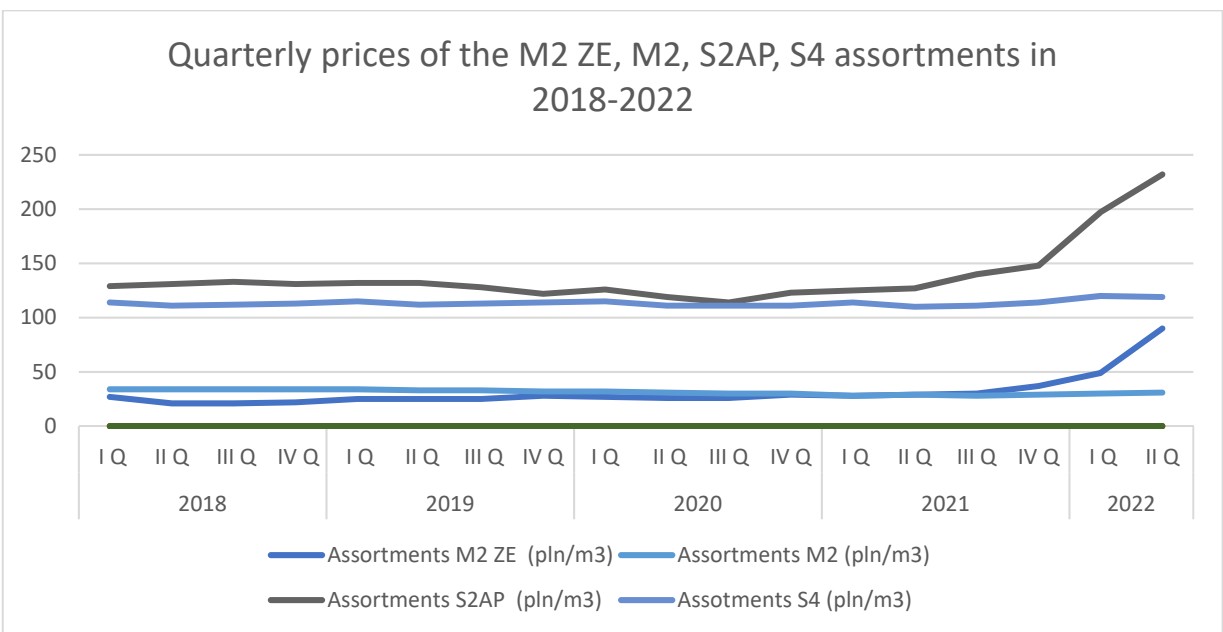

**Figure 1.** Quarterly prices for M2 ZE, M2, S2AP, S4 assortments in years 2018–2022. Source: Own elaboration based on data obtained from the State Forests IT System (own elaboration based on the Computer Management Support System of the State Forests National Forest Holding).

Raw timber prices are linked to vertical price transmission. However, due to the sales procedures carried out by the forest manager, taking into account periodic sales cycles, their impact is significantly delayed. The impact factors are price changes in global and regional markets [57–60].

It is important to remember that wood prices are the key variable in forest management optimisation and its forecasting is subject to a high degree of uncertainty [47]. Time series methods ensure good results while requiring modest input data, which makes them especially useful in the analysis of forest management issues [61]. The choice of the ARIMA method, as well as of the ARIMA method that takes account of seasonality (SARIMA), was based on a literature review in which the authors attempted to make predictions in various areas of the economy. We need to agree with Broz and Viego [62], who claim that using predictive techniques and models in the area of forestry is still rare. However, attempts were made to apply these methods in forestry. An article [63] by Greek researchers who investigated the possibility of using the ARIMA modelling for firewood attracted the most attention. The results of their research were satisfactory. Broz and Viego [62] applied the ARIMA method to analyse the prices of wood in Argentina. Japanese researchers have successfully applied the exponential smoothing (ETS) method and autoregressive integrated moving average (ARIMA) models to forecast monthly prices of logs obtained from the three most important tree species in Japan [48]. Soares et al. [64] demonstrated that ARIMA was the appropriate model for the forecasting of prices of eucalyptus wood. Yin [65] used one-dimensional ARIMA models to forecast prices in the USA and concluded that short-term forecasts were precise enough to be adopted in practice. Hetemäki et al. [2] used one-dimensional and multi-dimensional models of time series and concluded that they may serve as a useful tool for short-term price forecasts for Finnish wood [66]. Kolo and Tzanova [39] used time series analysis for short- and medium-term forecasting of the size of exports and imports of raw wood and of prices in the German wood market. Autoregressive modelling was used to analyse the predicted changes of pine (*Pinus sylvestris* L.) prices in the four main regions of Finland on the basis of actual monthly prices of stumps between January 1995 and June 2005 [51]. The artificial neural network model and the ARIMA model were also used to forecast the wear of pine lumber [67] and for the prediction of

the price of plywood [68] and cellulose, lumber export prices in Brazil [65], and for wood rubber price prediction [69,70].

Examples of other disciplines in which the ARIMA model was used include medicine [71–79]; the energy industry [80,81]; environmental sciences [82–91]; fire protection [92]; the stock market and crypto-currencies [93]; and the food industry [94,95].

## 3. Results

### 3.1. Forecast for M2 ZE Assortment (Residual Wood)

Prices of residual wood ranged between 20 and 30 PLN/m$^3$. We can observe a significant increase in the price in Q4 of 2021, which continued in Q1 and Q2 of 2022. This factor influences the forecasting procedure for the data provided (Figure 2).

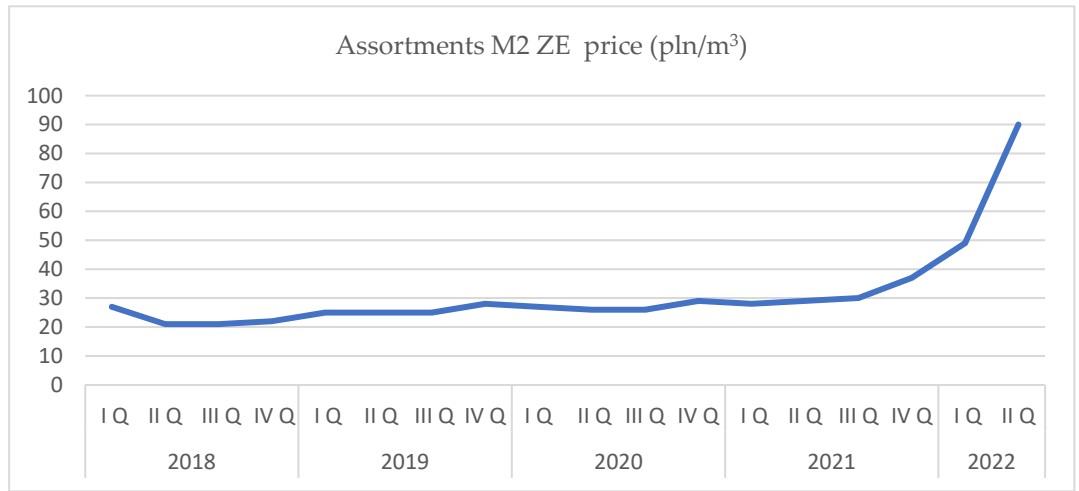

**Figure 2.** Prices for the M2 ZE assortment at quarterly intervals in years 2018–2022. Source: Own elaboration based on data obtained from the State Forests IT System (SILP).

The ARIMA (0, 2, 0) model was used during the data analysis to assign relevant *p, d, q* parameters to the ARIMA method (Figure 3).

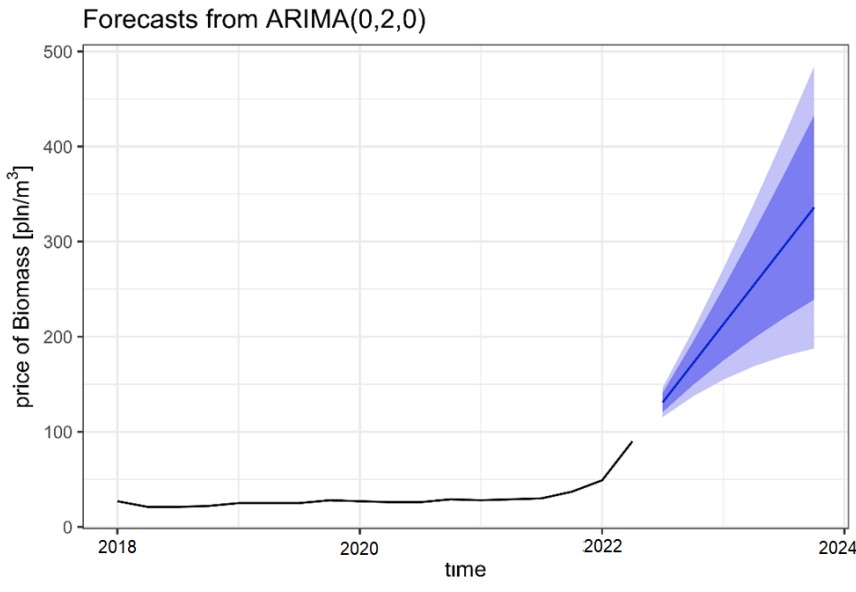

**Figure 3.** Price forecast for the M2 ZE assortment for Q3 of 2022, Q4 of 2022, and Q1 of 2023. Source: Own elaboration based on data obtained from the State Forests IT System (SILP).

The values of the point forecast for residual wood are on an upward trend. It is a significant hike in the price compared to 2021 (Table 1). Comparing the prices in the first two quarters of 2022, which increased by an extremely large amount, gives rise to the anticipation of a further increase in the following years.

**Table 1.** Price forecast for M2 ZE assortment for Q3 of 2022, Q4 of 2022, and Q1 of 2023. Source: Own elaboration based on data obtained from the State Forests IT System (SILP-own elaboration based on the Computer Management Support System of the State Forests National Forest Holding).

| Time | Point Forecast (pln/m$^3$) | Lo 80 * (pln/m$^3$) | Hi 80 * (pln/m$^3$) | Lo 95 * (pln/m$^3$) | Hi 95 * (pln/m$^3$) |
|---|---|---|---|---|---|
| 2022 Q3 | 131 | 120.83 | 141.17 | 115.45 | 146.55 |
| 2022 Q4 | 172 | 149.26 | 194.73 | 137.23 | 206.77 |
| 2023 Q1 | 213 | 174.95 | 251.04 | 154.82 | 271.18 |

* estimated value 80% and 95% confidence intervals.

### 3.2. Forecast for the M2 Assortment (Firewood Slash)

Price levels for firewood slash have not been subject to radical changes since 2018. A minor drop could be observed at the turn of Q4 of 2020 and Q1 of 2021. The trend line for the M2 assortment values can be regarded as stable (Figures 4 and 5).

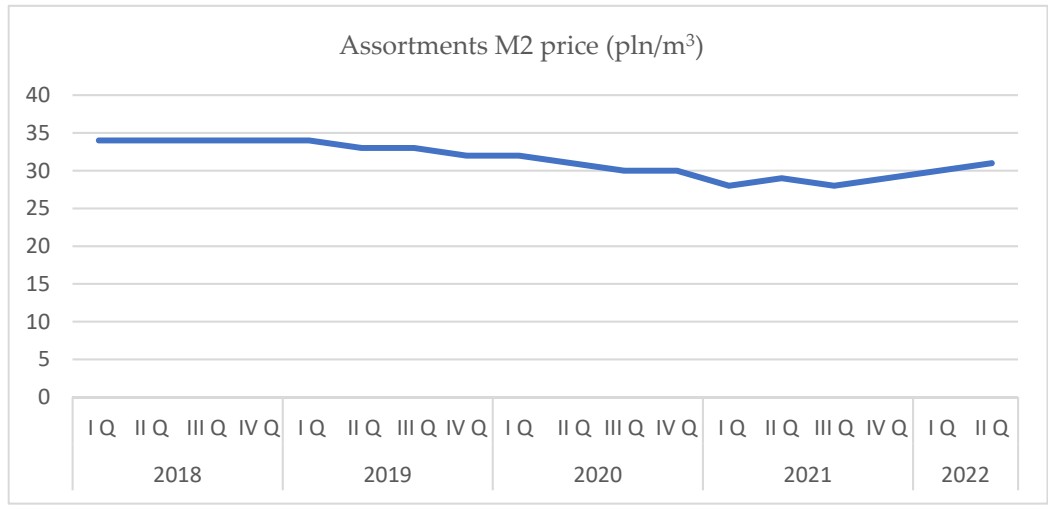

**Figure 4.** Prices for the M2 assortment at quarterly intervals in years 2018–2022. Source: Own elaboration based on data obtained from the State Forests IT System (SILP).

An analysis of the forecast for firewood slash indicates that prices remained at a stable level from quarter to quarter. Differences between quarterly prices ranged between PLN0.3 and PLN0.4 (Table 2).

**Table 2.** Price forecast for the M2 assortment in Q3 of 2022, Q4 of 2022, and Q1 of 2023. Source: Own elaboration based on data obtained from the State Forests IT System (SILP).

| Time | Point Forecast (pln/m$^3$) | Lo 80 * (pln/m$^3$) | Hi 80 * (pln/m$^3$) | Lo 95 * (pln/m$^3$) | Hi 95 * (pln/m$^3$) |
|---|---|---|---|---|---|
| 2022 Q3 | 31.79 | 30.96 | 32.61 | 30.52 | 33.05 |
| 2022 Q4 | 32.16 | 30.98 | 33.34 | 30.35 | 33.97 |
| 2023 Q1 | 32.14 | 30.16 | 34.11 | 29.12 | 35.15 |

* estimated value 80% and 95% confidence intervals.

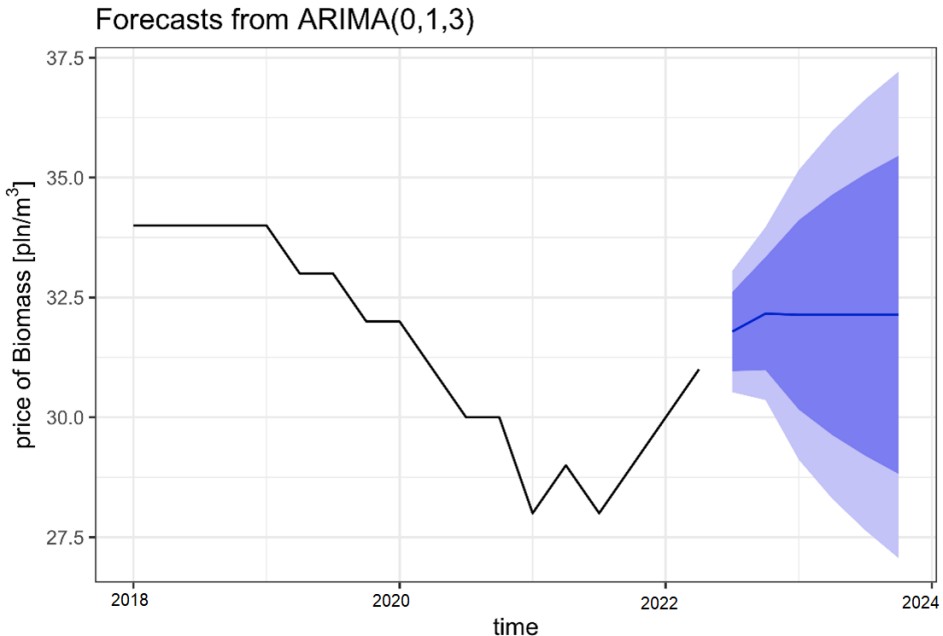

**Figure 5.** Price forecast for the M2 assortment in Q3 of 2022, Q4 of 2022, and Q1 of 2023. Source: Own elaboration based on data obtained from the State Forests IT System (SILP).

### 3.3. Forecast for the S2AP Assortment (General Purpose Cordwood)

Interpreting the graph above (Figure 6), one can observe an increase in the price of general-purpose cordwood that has continued since Q4 of 2021. A comparison of price levels in the period between Q1 of 2018 and Q2 of 2022 shows that the price increased by more than 100%. The most significant increase took place in Q4 of 2021 (Table 3).

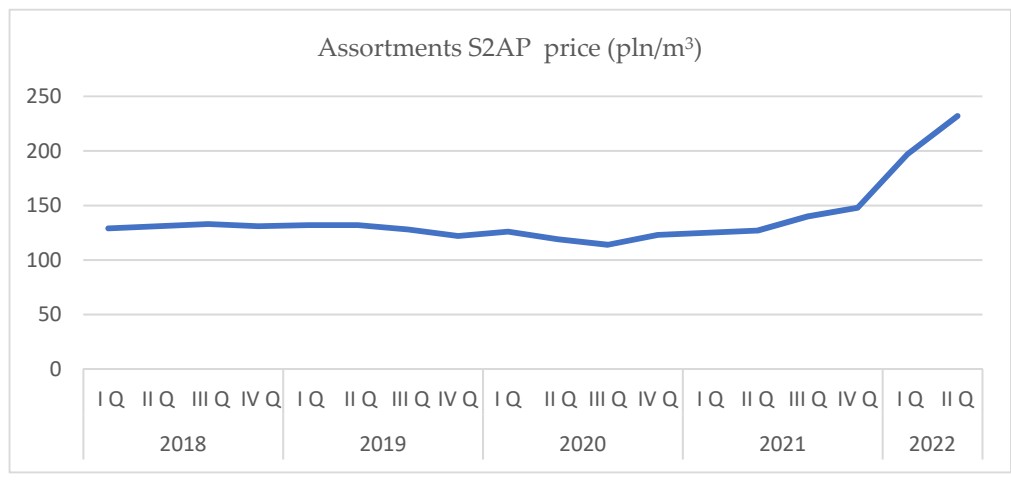

**Figure 6.** Prices for the S2AP assortment at quarterly intervals in years 2018–2022. Source: Own elaboration based on data obtained from the State Forests IT System (SILP).

**Table 3.** Price forecast for the S2AP assortment for Q3 of 2022, Q4 of 2022, and Q1 of 2023. Source: Own elaboration based on data obtained from the State Forests IT System (SILP).

| Time | Point Forecast (pln/m³) | Lo 80 (pln/m³) | Hi 80 (pln/m³) | Lo 95 (pln/m³) | Hi 95 (pln/m³) |
|---|---|---|---|---|---|
| 2022 Q3 | 272.93 | 257.9 | 287.96 | 249.95 | 295.91 |
| 2022 Q4 | 311.35 | 283.3 | 339.4 | 268.45 | 354.25 |
| 2023 Q1 | 350.83 | 305.94 | 395.72 | 282.18 | 419.48 |

The price forecast values for the S2AP assortment (Figure 7) are distinguished by an upward trend. Differences between the forecasts range by PLN40/m$^3$.

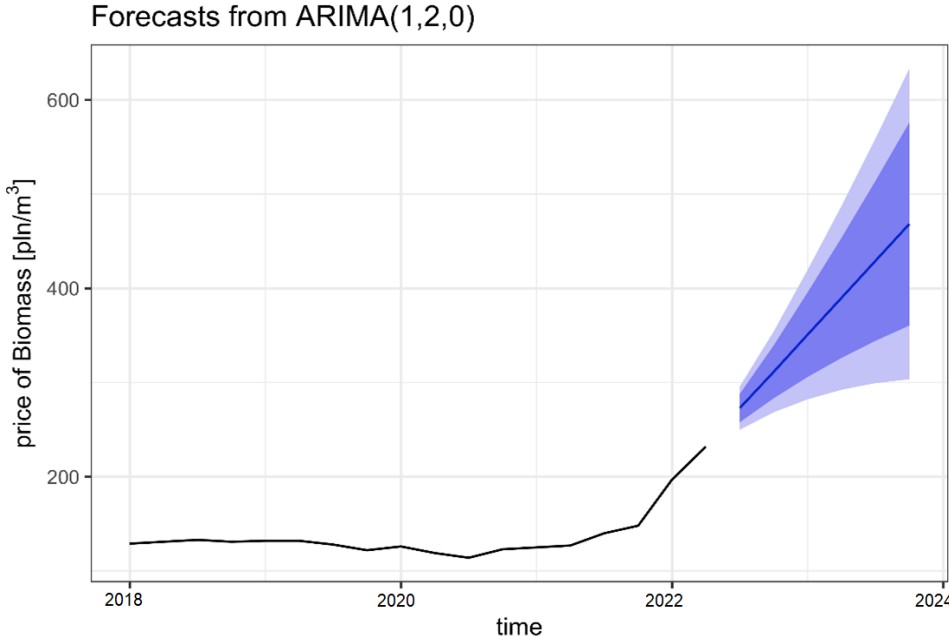

**Figure 7.** Price forecast for the S2AP assortment for Q3 of 2022, Q4 of 2022, and Q1 of 2023. Source: Own elaboration based on data obtained from the State Forests IT System (SILP).

### 3.4. Forecast for the S4 Assortment (Firewood)

The analysis of firewood price values has a seasonal characteristic (Figures 8 and 9). The increase in price levels takes place in quarters 4 and 1 of the subsequent years. That period covers the autumn and winter months, in which an increased demand for firewood is observed. However, attention needs to be paid to the significant price increase in Q3 of 2021 and to the fact that this situation continued in Q1 of 2022. The difference between the highest price noted in Q1 of 2021 and in Q1 of 2022 amounts to 5%, which significantly exceeds the difference between previous years.

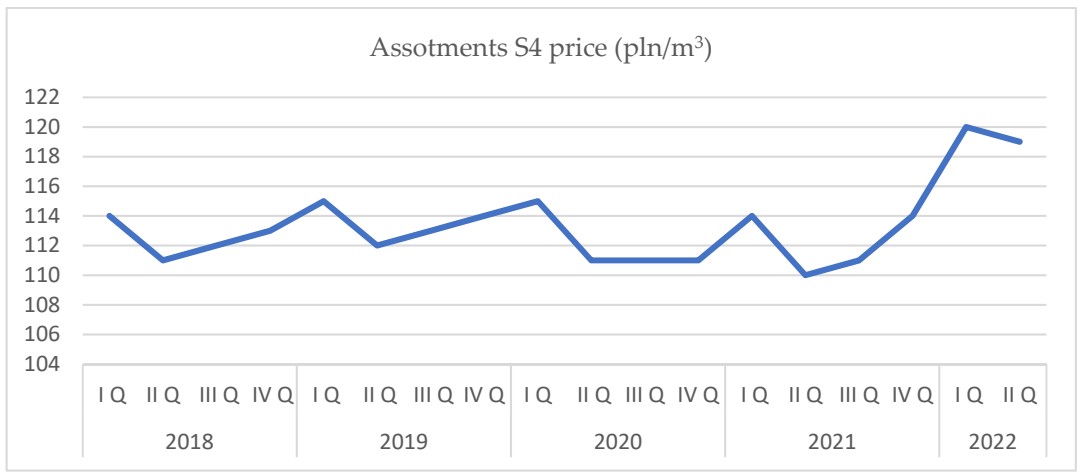

**Figure 8.** Prices for the S4 assortment at quarterly intervals in years 2018–2022. Source: Own elaboration based on data obtained from the State Forests IT System (SILP).

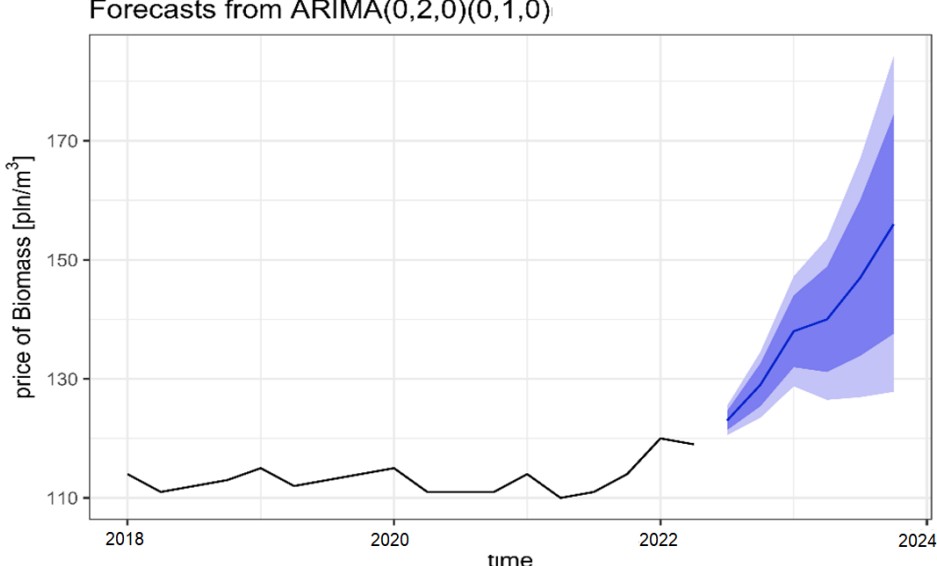

**Figure 9.** Price forecast for the S4 assortment in Q3 of 2022, Q4 of 2022, and Q1 of 2023. Source: Own elaboration based on data obtained from the State Forests IT System (SILP) [4].

The characteristic of seasonality for the S4 firewood was established through data analysis (Table 4). Consequently, the SARIMA model, expanded by the seasonality factor, was applied.

**Table 4.** Price forecast for the S4 assortment in Q3 of 2022, Q4 of 2022, and Q1 of 2023. Source: Own elaboration based on data obtained from the State Forests IT System (SILP).

| Time | Point Forecast (pln/m³) | Lo 80 (pln/m³) | Hi 80 (pln/m³) | Lo 95 (pln/m³) | Hi 95 (pln/m³) |
|---|---|---|---|---|---|
| 2022 Q3 | 123 | 121.38 | 124.62 | 120.53 | 125.47 |
| 2022 Q4 | 129 | 125.39 | 132.61 | 123.47 | 134.53 |
| 2023 Q1 | 138 | 131.95 | 144.06 | 128.75 | 147.25 |

For the purpose of the forecast sensitivity test, actual prices for the third quarter of 2022 were taken into account, as this information was reported by PGL LP. The remaining periods are not finished, which makes it impossible to provide final data.

The results of the sensitivity test for the use of the ARIMA model to forecast forest biomass prices are presented above. The largest errors are characterised by the M2ZE assortments of approximately 45.56% of the percentage error, while the smallest error was found in the M2 assortment, where MAPE was 2.55% (Table 5).

**Table 5.** MAE, MAPE, RMSE results for Q3 2022. Source: Own elaboration based on data obtained from the State Forests IT System (SILP).

| Assortments | M2ZE (pln/m³) | M2 (pln/m³) | S2AP (pln/m³) | S4 (pln/m³) |
|---|---|---|---|---|
| 2022 IIIQ real | 90 | 31 | 232 | 119 |
| 2022 IIIQ forecast | 131 | 31.79 | 272.93 | 123 |
| MAE | 41 | 0.79 | 40.93 | 4 |
| MAPE [%] | 45.56 | 2.55 | 17.64 | 3.36 |
| RMSE | 41 | 0.79 | 40.93 | 4 |

## 4. Discussion

The forms of trade in wood raw material are based on market prices of the sourced assortments, which are influenced by multiple factors. The global climate change that is being observed, the adjustment of regulations, and the increased logging caused by biotic and abiotic factors are the main factors influencing fluctuations in the wood trade in the last decade. The largest logging incidents are mainly related to the significant number of climate catastrophes causing damage to trees and an increased supply of roundwood on the market. In most cases, this fact resulted in a drop in wood prices or their partial stabilisation. In the future, the wood utilisation sector will be the most stable market, as the interest in and consumption of renewable sources of renewable biomass is on an upward trend. According to researchers, the EU's policies [6] contribute to that [96]. Due to these changes, there is currently strong competition in the firewood and industrial wood markets (pulp and paper industry), which leads to structural changes in the supply of wood. It is one of the key elements that will be influencing demand among wood producers in the near future.

There are specific factors that influence the wood trade in Poland. Price fluctuations on the market are distorted by the single dominating entity, which is PGL LP (responsible for 80% of managed forests in Poland), while all minor entities have to adjust their price strategies to the dominating entity. Price levels in the country depend on local supplies and economic needs on the market. Local influences, the structure of wood processing companies, and the specific approach to wood trade in Poland influence the structure of the wood assortment supplied to the industry. This is confirmed by the increase in the volume of wood delivered to the consumer, also in the case of the lowest quality assortments, in recent years. This has necessitated a change in the perception of the problem of wood price forecasting. Solving the problems of the wood market can be based on the monitoring of the development of the market of wood consumers in the country, as well as on the European and global markets, on the basis of the assessment of wood trade levels; making trade agreements; conducting assessments at the appropriate frequency; and reacting to these changes. Due to the high frequency in price changes at this point, we can opt for a short-term forecast, as the price level is also affected by the political and economic situation in the country. We are unable to predict correctly what prices will be in mid-2023, as external factors may change drastically. Therefore, the authors undertook to forecast only quarterly series in the near term. The inflation aspect was explained above.

Predictive analyses of price fluctuations have to focus on new sectors in forestry (carbon sequestration, carbon trading, renewable sources of energy). Wood processing plants have to concentrate on developing new products based on wood and on investing in wood processing technologies [97]. The wood industry exhibits a certain supply cyclicity [98] which results from complex interactions between market factors that determine the situation in the primary wood market (with regard to demand, prices, and supply of wood). Differences in the cyclicity of price fluctuations between different industries result largely from the durability of the product, as durable goods industries are several times more cyclic than non-durable goods industries [99]. It is also suggested that technological progress and changes in consumers' preferences determine long-term trends in the forest products sector [100]. Cyclic price patterns in the Polish market were largely connected with the condition of the national, European, and global economy and constituted the main factor responsible for general wood price fluctuations. The years 2018–2022 were a period of dynamic economic changes caused by an expansion in the economic development of the timber industry (in 2020). In addition to the increased demand for raw material from European Union countries, the increase in the supply of raw material in North America was significant. During the pandemic and the increase in demand for lumber in North America, the major lumber processors in Europe increased their lumber exports. European prices increased significantly due to the shortage of sawn materials, which translated into increased demand for roundwood [60,101].

Cyclic fluctuations can be observed that revealed large amplitudes in the real prices of wood in Poland, reaching PLN103/m$^3$ for the S2AP assortment in the years 2018–2022 and PLN63/m$^3$ for the M2ZE assortment. After two years of significant price increases, the collapse in demand translated into a drop in prices from Q1 of 2020, with a shift occurring in Q3 of 2021. The turning points for prices coincided with turning points for the European wood market. An upward trend in the real prices of wood was observed in the years 2018–2022. The situation in the European wood market was also linked to the phases in the global economic cycle. The drop in wood prices in European countries in 2019 was caused by the health crisis triggered by COVID-19. The construction and renovation increases are related to government stimulus packages following the COVID pandemic, as well as to the low bank interest rates at the time. Q1 and Q4 of 2019 brought a drop in sales. Cyclic price fluctuations coincided with the peak in the Polish wood economy (2021/2022) due to the growing demand from industry, especially construction. The growth in construction and renovation was linked to the government's stimulus packages after the COVID pandemic, as well as low bank interest rates at the time. Similar to 2019, supply diminished in Q1 of 2021. However, in Poland, demand for wood assortments does not always reflect trends in GDP [12]. The decline in demand for wood products was seen during periods of slower economic growth. This led to a decrease in timber harvesting. This consequently caused fluctuations in supply and increases in demand, which coincided with gradual cycles of price changes.

The price of wood derived from its supply (in terms of amount as well as assortment structure) depends primarily on the degree of implementation of forest management plans. Despite the volatility in global markets and the significant drop in demand in 2019, the State Forests managed to fulfil its sales plan, albeit at slightly reduced prices. Research [102] confirmed the high price elasticity for the supply of small- and medium-sized wood in Europe [103]. According to research conducted in Europe and on other continents, the price and supply of wood are shaped by market and non-market factors, depending on the products and the geographical region [104]. Industrial wood (M2, M2ZE, S2AP, and S4) displays a lower price elasticity than large-sized wood [105], which suggests that price fluctuations have less impact on the supply of pulpwood (industrial wood) than on sawmill wood. Taking into account the biomass assortments presented in the publication, which in 2021 were recognised as energy wood by the Director General of the State Forests, the quality and dimensional classification applicable in Poland was followed. This is due to, i.e., the situation in the US market, which resulted in higher prices of sawn timber and caused a rise in the prices of large-sized wood in a short period of time [106,107]. Analyses conducted with the application of predictive models [39,47,61,62] and, in the case analysed here, with the application of the popular ARIMA model [48,86,108], confirmed the occurrence of the predicted increases in the price of wood on the general level for the M2ZE and S2AP assortments by 45% and 17%, respectively, between Q2 of 2022 and Q3 of 2022 (forecast). The increase for the remaining assortments M2 and S4 was not significant (2%; 3%). The most significant increase in real prices was observed for the S2AP and M2ZE assortments, for which a major price hike could be observed at the turn of Q1 and Q2 of 2022. The prices of middle-sized wood peaked in Q3 of 2022. This ensures predictable revenues, at the same time restricting the possibility of making economically beneficial decisions during the year. Analysing information with the application of autocorrelation of the prices of wood raw material will make it possible to illustrate the seasonality of assortments. This situation can be observed with regard to the S4 assortment, which achieved greater popularity in the winter months (October–December). This applies to the price increase observed in that period. It also increases the price of fuel products for individual customers buying wood from private companies in the wood industry. The question arising at this point is what other wood assortments are seasonal as well.

The more stable the economy and the better the economic indicators, the lower the frequency and amplitude of wood price fluctuations. Natural disasters (such as windfalls or disease affecting spruce and pine stands) and other unpredictable events were local

and did not impact the supply of wood on a national scale, with the exception of spruce stands in the years 2017–2018 (degraded by the bark beetle) [109]. It can also be concluded that changes in local prices in degraded areas did not have a significant impact on the price of wood in Poland overall. Sourcing wood from stands damaged by the hurricane, which affected northwest Poland in August 2017, resulted in an additional supply of approximately 3 million m$^3$ of wood, also in 2018 (the annual harvest of wood in Poland is approximately 39 million m$^3$) [110].

Price increases can be observed across Europe; however, comparing the predicted increase in wood prices between countries is complicated due to different quality requirements for definitions of assortment classes [111]. However, these can be simplified, using the M2, M2ZE, S2A, and firewood S4 assortments applied here [41]. It is important to note that prices of wood in neighbouring countries in Central Europe were similar (which suggests that there was room for a further increase in the Polish market). As stated earlier in the chapter, timber pricing in Poland is specific and includes periodic information on the inclusion of harvesting, skidding, and administration costs in the minimum price. If one of these factors increases, the price will also increase. The setting of the maximum price by the State Forests in previous years was intended to maintain the stability of the timber industry in Poland.

The high frequency of natural disasters in the whole of Central Europe (windfalls) had a negative impact on the prices of coniferous wood. There was an oversupply of spruce wood in Southern Poland. As prices in Poland were relatively low compared to Austria, transportation costs were acceptable, and currency exchange rates favourable, domestic and foreign companies showed more interest in the product, which led to price reductions [15]. Prices were lowered for Polish entrepreneurs who were the main recipients of wood, and such large fluctuations that have taken place recently have affected the overall price of wood raw material. For companies in the wood industry, these prices were high, which resulted in the lack of profitability of doing business. Such decisions have huge effects in the Polish economy, where the wood industry plays an important role. The indication S2A applies to wood in Poland, which, being already damaged, is not subject to such a strong decline in value as other roundwood sorts. Research conducted in Japan showed a correlation between lower monthly prices between June and August due to damage caused by pests [112]. Similar activities can be seen in Europe; however, due to the assumptions of sustainable development, European procedures are longer and not so strongly exposed. However, attention is included, and the impact of degradation in European countries is indicated [113]. Assortments with a medium diameter (S2A) showed greater price stability. No strong fluctuations in the prices of that assortment were observed; however, an evenly increasing price trend can be detected in the period investigated (2018–2022), which is due to the continuity of harvests. However, price forecasts point to significant increases in the value of that assortment.

The comparison of data with the application of the ARIMA model showed a significant increase in the prices that may be adopted by the State Forests in 2023, and in subsequent years, that is due to, i.e., the change in prices between the Polish and Western European markets on the one hand, and the Baltic and Scandinavian markets on the other [31].

In the face of the energy crisis in Central Europe, new solutions, relying on renewable sources of energy, are being explored. Measures intended to ensure access to heat and electricity are among the main considerations of specialists in the field. Assortments that were subjected to the analysis and forecasting can be used in the production of heat (as forest biomass), which relieves the burden of coal use. This adheres to the legislative documents of the Green New Deal [6], which defines the common climate and environment protection policy. The use of forest biomass as a renewable source of energy enables the reduction of the carbon footprint generated by the heat production industry. Access to real-time information about the price of the resource is highly significant for businesses operating in the wood industry and for private consumers who need to plan their short-term budgets. Attention should also be paid to the situation to the east of Poland. Among

the main suppliers of wood residues were countries such as Ukraine, Russia, and Belarus. Due to the war situation across Poland's eastern border, supplies have been significantly restricted. The reduction in resources due to the discontinuation of imports from the east has also been the cause of the increase in the prices of wood raw materials destined for biomass. The import of pellets from Ukraine is of great importance to Poland. Our country was the largest importer of this product—in 2021 we imported a total of 121,000 tonnes from Ukraine, which accounted for nearly 30% of all Ukrainian exports (412,000 tonnes). Considering that the volume of pellet production oscillates around 1 million tonnes, this is quite a serious loss [114].

## 5. Conclusions

Wood prices show a significant upward trend over time, both in terms of direction and the broad amplitude of change. The verifiable temporal distribution was used to analyse the overall price volatility of wood raw material. The practical aspect of the results of the research consisted in identifying seasonal fluctuations for small- and medium-sized wood assortments as factors influencing the situation on the wood market.

Strong deviations in the predicted price levels occurring over short time periods (1–2 quarters) are often caused by random events that are difficult to identify. In the time period studied (2018–2023), irregular fluctuations in the predicted price changes for small- and medium-sized assortments constituted a small part of the analysed data. The local character of the occurrence of random distortions did not have a significant impact on the price of these assortments.

Seasonal fluctuations in the prices of S2A, M2, M2 ZE, and S4 wood occurring over the period of one year are related to seasons, weather conditions, climate change, and seasonality in the sourcing of wood and demand for wood. The S4 assortment (firewood) displayed seasonality too. Its price increased in Q4 of each year. No seasonality was observed for other assortments. Price patterns depend on the assortment of wood; the lowest predicted price increase value was observed for the M2 assortment, while the highest value was observed for the S2AP assortments. The supply of hardwood was the highest in Q1 and lowest in Q3.

The analysis of the overall price volatility and wood supply enables the application of ARIMA models for the forecasting of wood prices and planning the amounts of wood to be sourced in longer time periods, as well as for the adjustment of the wood assortment structure to the changes in market demand.

Long-term price fluctuations displayed a minor upward trend, while the predicted price of wood over the last 2 years was distinguished by a clearer upward tendency, which continued over the forecasting period (the percentage increase in the prices of M2ZE and S2AP assortments amounted to 45% and 17%, respectively). In the long term, the growing demand for medium-sized wood will cause a rise in prices as a result of oversupply.

Price forecasting is the result of complex mechanisms. The ARIMA model captures past market behaviour using a single data series. This low data requirement is a strength of the ARIMA approach in a forecasting situation with many unknowns.

The weakness of the model is its reliance on a single time series of prices from forecasting. It does not allow for the inclusion of future changes caused by exogenous factors: transportation prices, trade restrictions, natural disasters, etc.

Forecasts are useful for creating a short-term forecast, one or two quarters; for use beyond this period, other methods are needed that take into account exogenous factors.

Due to the large number of exogenous factors affecting the forecast of wood prices in Poland, the analysis is difficult. In the case of natural disasters, the political and economic situation in Poland, the authors assumed in their study that the price includes this information. This also applies to the level of inflation in the country. In their further plans, the authors want to attempt to expand the forecasting model with external factors influencing the determination of the level of wood prices in Poland.

**Author Contributions:** Conceptualisation, A.G. and K.A.; methodology, A.G.; software, A.S.-B.; validation, K.A., Z.S. and M.W.; formal analysis, Z.S.; investigation, A.G.; resources, A.G.; data curation, M.W.; writing—original draft preparation, A.G.; writing—review and editing, M.W.; visualisation, A.S.-B.; supervision, K.A.; project administration, M.W.; funding acquisition, K.A. All authors have read and agreed to the published version of the manuscript.

**Funding:** This research was funded by the National Centre for Research and Development, BIOSTRATEG3/344303/14/NCBR/201 and POIR 01.01.01-00-802/19. Publication was financed within the framework of the Polish Ministry of Science and Higher Education's program: "Regional Excellence Initiative" in the years 2019–2022, project no. 005/RID/2018/19, financing amount 1,200,000,000 PLN.

**Data Availability Statement:** Not applicable.

**Conflicts of Interest:** The authors declare no conflict of interest.

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
