# Peer review of "Biomass Price Prediction Based on the Example of Poland"

_forests, doi:10.3390/f13122179_

Round 1
Reviewer 1 Report
- Use scatter with line graph to show the price changes.
- Did the price was deflated using consumer price index? I suggest explaining it in the method section.
- The details of stationary test and ACF and PACF graphs can be included in the paper to provide more information.
- The quality of forecasted model by ARIMA is not mentioned in the text. In the Did you have the valuses of root mean square, mean absolute error, mean absolute percentage error and the Theil Inequality Coefficient?
- Language editing is required.
Author Response
Reviewer 1
Thank You very much for your thorough evaluation of our publication. Your comments and corrections are very valuable. They represent a significant improvement in the quality of the publication. We hope that the present explanations will be satisfactory to You.
With best regards
Authors
Comments and suggestions for Authors
Use scatter with line graph to show the price changes.
1 Thank you for the hint regarding the chart. It will be amended.
Corrected
Did the price was deflated using consumer price index? I suggest explaining it in the method section.
Inflation was included in the prices subjected to the forecast analysis. This has been added to the article.
At the stage of developing methodological assumptions, the authors considered including the inflation rate in the source materials used to prepare forecasts. Taking into account the fact that the forecasts generated using the proposed method are to be used for utilitarian purposes, it was concluded that the inflation factor is a component of the analyzed prices. In our research, we adopted the definition that the price is a numerical expression of the value of money that should be spent on the purchase of a given good at a given time. So we're predicting a nominal value that takes inflation into account. It should be emphasized that the article is the beginning of the search for appropriate methods of predicting price changes on the biomass market and the obtained results are to be used in the future as the basis for further research. In the series of subsequent articles, we will also carry out research taking into account the real price. Using the material presented in this article, we will obtain a comparative base as a source material. We hope that the source data and forecast results presented in this form will also serve other researchers as a reference point for further analyses. Summing up, we would like to thank the reviewer for drawing attention to the inflation factor, which will be taken into account when preparing source materials for subsequent studies related to the analysis of time series.
The details of stationary test and ACF and PACF graphs can be included in the paper to provide more information.
Stationarity test of ACF and PACF was done to each sort position to be able to determine specific variants of ARIMA methodology, e.g. ARIMA (0,2,0). Unfortunately, the too large size of the graphs would technically prevent transparency of the publication.
We also enclose a description of the steps of the conducted research. Including this in the article would technically not fit in the article due to the large size of the information.
- The quality of forecasted model by ARIMA is not mentioned in the
The quality of forecasted model by ARIMA is not mentioned in the text. In the Did you have the valuses of root mean square, mean absolute error, mean absolute percentage error and the Theil Inequality Coefficient?
Corrected in text
The sensitivity analysis was performed in comparison with the third quarter of 2022, as this period can be considered closed and PGL LP reported the data.
Table 5. MAE, MAPE, RMSE results for Q3 2022. Source: Own elaboration based on data obtained from the State Forests IT System (SILP).
Assortments |
M2ZE [pln/m3] |
M2 [pln/m3] |
S2AP [pln/m3] |
S4 [pln/m3] |
2022 IIIQ real |
90 |
31 |
232 |
119 |
2022 IIIQ forecast |
131 |
31,79 |
272,93 |
123 |
MAE |
41 |
0,79 |
40,93 |
4 |
MAPE [%] |
0,455556 |
0,025484 |
0,176422 |
0,033613 |
RMSE |
41 |
0,79 |
40,93 |
4 |
The results of the sensitivity test for the use of the ARIMA model to forecast forest bi-omass prices are presented above. The largest errors are characterized by the M2ZE assortments of about 0.41% of the relative percentage error, while the smallest error was found in the M2 assortment, where MAPE was 0.025%.
Language editing is required.
Thank you very much for attention. Language errors will be corrected.
Corrected
The comments have been taken into account and, within the framework of available data, incorporated into the article's revision.
We thank the Reviewer for important comments that enhance the work.
The suggestions are very pertinent and will help in the development of future articles.

Reviewer 2 Report
The beginning of the article provides a pleasant introduction to the various issues
impacting the polish forest sector and the discussion sections provides a lot of
interesting insights.
My main concern is that the forecasting model is sensitive to the large price increases
in recent months. The article would benefit from a sensitivity analysis on this aspect.
This would require splitting the data into a training and a test set and comparing the
output of your prediction on the real test data, more details further down below. The
small number of observations (18) is also a concern for the validity of the approach. It
would be useful to have time series data that represents several economic cycles.
The authors should make a greater effort at taking the influence of the rest of the
economy into account. In particular, the forecasting model could be improved with the
following information on the past (training) side of the observations:
1. correcting price series for the effect of inflation
2. controlling for the effect of natural disturbances such as storms and insect damages
followed by salvage logging
3. looking at price transmission with the global prices of fuel wood and roundwood
(using unit prices of trade from a global trade database)
The forecasting model could also be improved on the future (prediction) side, by feeding
it exogenous variables, such as:
a. forecasts of crude oil or natural gas prices
b. forecasts of pulp prices
It would be a lot of work to add all the point above, so I will not insist on it. I
strongly encourage you to preform the sensitivity analysis and the inflation correction.
These methods are within reach and should be added to the article. The other points are
more difficult, but considering the very clear influence of exogenous drivers, the
author should at least describe the limits of their forecasting model.
I don't think the article can be considered as a proof of the effectiveness of the
forecasting as written at the end of the abstract lines 31 to 33:
> "Based on the research we have conducted, we proved that forecasting the prices of the
> analysed assortments using the ARIMA and SARIMA models is effective."
Considering the high uncertainty, I think the word "proof" is too strong here. A first
attempt of illustrating the fitness of the model would be to perform the sensitivity
analysis requested below.
Lines 80, 82
> "In recent years, the share of small wood M2 and medium-sized wood S2A assortments
> (25%) in the total amount of wood sold by PGL LP has been on the increase [20].
Could you specify why there was an increase of small an medium size wood assortments in
Poland?
Could there be an influence of the salvage logging related to bark beetle?
You mention salvage logging later lines 365-368:
> "Natural disasters [...] were local and did not impact the supply of wood on a
> national scale, with the exception of spruce stands in the years 2017-2018 (degraded
> by bark beetle)"
Lines 85, 86
> Adjusting prices to the level of supply can be considered for optmising the benefits
> of price fluctuations [23,26].
In economic theory, price is the result of a market equilibrium between market
participant. In such models, it is assumed that consumers have different willingness to
pay and producers different willingness to accept. The demand (or supply) quantity at
any given price draws a demand (or supply) curve. The crossing point of the demand and
supply curves represents the equilibrium price. You briefly mention such studies lines
96, 97:
> "most studies being based on models of supply and demand in various geographical
> regions"
Since there is only one large supplier in Poland, the "State Forests - National Forest
Holding" would you describe this situation as a monopoly, or a quasi monopoly? In this
situation price setting is not done through a market equilibrium but through some other
mechanism. Could you please provide more details on the mechanism for adjusting prices
in that context?
You use a forecasting method based purely on past price data, it is important to make
sure that the price setting mechanism didn't change throughout the period. Do you have
indication that this was the case?
Related to this, you wrote about market distortion lines 289, 292:
> "Price fluctuations on the market are distorted by the single dominating entity, which
> is PGLLP (responsible for 80% of managed forests in Poland), while all minor entities
> have to adjust their price strat- egies to the dominating entity."
Lines 78, 80
> "decisions of the Director General of the State Forests [20], which regulate the rules
> for signing contracts with enterprises as part of the so-called "purchase history" and
> free access to purchases at higher price levels"
It seems like wood buyers from the "purchase history" contracts can buy at a
preferential, lower price. Do you distinguish low level from high level purchases in
your analysis? What is the proportion of sales in the two markets ("purchase history"
contracts vs high price level purchases? As prices increased, has the proportion of high
level purchases changed throughout the study period?
It seems this market has two regimes, a fixed price regime and a free floating price
regime. Could you characterize them a little more?
The figure 1 says "Source: Own elaboration based on data obtained from the State Forests
IT System (SILP)". What level of details do you have in that source? Can you give the
number of observations per assortment?
Would it be possible to distinguish and separate these "purchase history" and free
floating segment of the markets in your analysis?
You could mention the literature on vertical price transmission (or lack thereof). That
literature is concerned with the study of how price increases in the intermediate or
final product sectors (sawnwood, panel, paper), get transmitted by back to the industry
and then transmitted back to the roundwood prices.
Line 119
Please explain the meaning of alpha_t, z_t, a_t,
It is a good practice to give an equation number.
Line 122
> "Is a function related to the non-seasonal autoregressive parameter p."
Add "and where" to introduce the next equation. Suggestion:
Is a function related to the non-seasonal autoregressive parameter p and where
Line 130
Please explain "wraz"
Lines 153, 154
> "The data include quarterly data for the period 2018 - second quarter of 2022, which
> provides exactly 18 observations for the forecast."
18 observations is a very short time series for forecasting. Could you describe how the
small sample size is likely to affect your result?
Is it possible to use pre-2018 data? How far could you go back in time with the data
source used? Please perform the analysis with a longer time series.
Lines 156, 157
> "Figure 1. Quarterly prices for M2 ZE, M2, S2AP, S4 assortments"
Using lines instead of bars for the price chart would make it easier to see price
trends.
Lines 160, 162
> "Time series methods ensure good results while requiring modest input data, which
> makes them especially useful in the analysis of forest management issues [57]."
For sure it is easier to use only "modest input data" to forecast future wood prices,
but wood prices are also influenced by the rest of the economy as you wrote lines 54, 55
and in the discussion section. To capture these influences in your model, you could add
exogenous variables. In particular - since you focus on fuel wood - it would be
interesting to look at price forecasts of crude oil or natural gas, as they are
potential substitutes for biomass use.
Please explain if possible how you would add exogenous variables to your model and
explore a little bit the literature linking biomass prices to fossil fuel prices.
Lines 196, 198
> "Prices of residual wood ranged between 20 and 30 PLN/m3. We can observe a signif-
> icant increase in the price in Q4 of 2021, which continued in Q1 and Q2 of 2022. This
> factor influences the forecasting procedure for the data provided."
As explained in "An introduction to statistical learning: with applications in R"
by G James, D Witten, T Hastie, R Tibshirani - 2013, could you please split your input
data into a training set and a test set? This would enable you to test the validity of
your forecast against real data. Please perform the ARIMA estimations by removing more
and more data from the training set:
1. removing the Q4 of 2021
2. removing also the full data from 2021
3. removing also the data from 2020 and 2021
Then generate the price forecasts for the M2ZE, M2, S2AP and S4 assortments again and
provide these as additional plots with doted lines or some way to distinguish the 3
removing steps above. You may leave these plots as annexes to illustrate the effect of
recent price hikes on the increased prediction.
This sensitivity analysis is also related to my previous comment on the length of the
time series. If you can get more data before 2018, then this would be better. It is also
important to consider the number of fixed contracts "" versus free floating contract
throughout the period.
Lines 210, 2013
> "The values of the Point Forecast for residual wood are on an upward trend. It is a
> significant hike in the price rise compared to 2021. Comparing the prices in the first
> two quarters of 2022, which increased by an extremely large amount, gives rise to the
> antici- pation of a further increase in the following years."
How likely it is that the price rise will continue? Could it be that prices revert to
some lower levels. You may also add that uncertainty increases with time i.e. the
forecast for Q1 of 2023 is much more uncertain than the one for Q3 of 2022. By the way
Q3 2022 is real data now, how easy is it for you to update this data point?
It would also be important to compensate for inflation. Please use a measure of national
inflation to correct the price series for the effect of inflation. If the effect is
small, you may remove it in the final analysis, but it is important to take this into
account.
Figures 2,4,6,8
You could simplify the x axis by removing the repeated years and the "kw" symbol (what
does kw mean?).
Figures 3,5,7,9
Displaying all years on the x axis would make it easier to read. In general the x axis
of all graph could be harmonized.
Lines 269
> "the SARIMA model, expanded by the seasonality factor, was applied"
Describe how the SARIMA model differs from ARIMA in terms of additional variables in the
equations lines 118 to 129.
Lines 276, 279
> "The global climate change that is being observed, the adjustment of regulations, and
> the increased logging caused by biotic and abiotic factors, are the main factors
> influencing fluctuations in wood trade in the last decade."
Modelling the increase of these external influences would require introducing exogenous
variables wouldn't it?
Lines 284, 286
> "Due to these changes, there is currently strong competition in the firewood and
> industrial wood markets (pulp and paper industry), which leads to structural changes
> in the supply of wood."
Since pulp markets are globalized - and considering there is some level of transmission
from pulp prices to roundwood prices - pulp forecasts are another candidate for an
exogenous variable that could drive your price forecast.
Lines 298 to 302
> "monitoring of the development of the market of wood consumers in the country, as well
> as on the European and global markets, on the basis of the assessment of wood trade
> levels; making trade agreements; conducting assessments at the appropriate frequency,
> and reacting to these changes."
Consider whether the notions of vertical price transmission and horizontal price
transmissions would improve the way you describe the reactions of the polish wood price
markets.
Lines 313, 315,
> "Cyclic price patterns in the Polish market were largely connected with the condition
> of the national, European, and global economy, and constituted the main factor
> responsible for general wood price fluctuations."
If this is the case, then you should try hard to get a longer time series. What is the
length of the economic cycle in the literature you cite? It would be useful to have
time series data that represents several economic cycles, at least two or more cycles.
Longer term data would train your ARIMA model to be better at predicting the likely
decrease of prices, once the economy moves into a new phase of the economic cycle.
Lines 317, 318
> "to the increase in the supply of raw materials in North America"
I don't understand, is this an increase of wood exports from Poland to North America, or
an increase of imports from North America into Poland? Please clarify.
Lines 324, 325
> "A cycle was identified for the analysed period of 3 to about 4 years."
How was the cycle length identified? The price series are just 4 years long from 2018 to
2022. This calls again for a time series that is several times longer than the economic
cycle.
Lines 329, 3030
> "due to the growing demand from industry, especially construction."
You could add that construction and renovation increases are related to government
stimulus packages following the COVID pandemic as well as to the low bank interest rates
at the time.
Lines 331, 334,
> "as a downward trend in wood sales has been historically observed in the period of
> slower economic growth too. Usually, the drop in supply would coincide with upward
> price cycles."
Not very clear, is this a drop in supply due to a supply constraints? Does demand remain
constant in that context? Try to improve.
Lines 337, 338
> "Due to the volatility in global markets and the significant drop in demand in 2019,
> the State Forests managed to fulfil its sales plan, albeit at slightly reduced
> prices."
A verb is missing from this sentence. Or maybe you should start the sentence with
"despite" instead of "Due to".
Lines 342
> "Industrial wood (M2, M2ZE, S2AP, and S4)"
The definitions given lines 147, 150 don't correspond to the FAO definition of
industrial roundwood. M2, M2ZE, S2AP, and S4 are mostly fuel wood products aren't they?
See the FAOSTAT Forest Product Production Statistics – Data Structure
> "residual wood M2ZE, firewood slash M2, medium-sized firewood S4 with the permissible
> level of soft rot set at 50%, minimum top diameter of 5 cm without bark, and
> general-purpose cord- wood S2AP"
It seems these assortments correspond to fuel wood types. Please clarify or explain why
you put them under industrial roundwood.
Lines 371, 372
> "resulted in an additional supply of about 3 million m3 of wood, also in 2018"
Please add the overall Polish wood supply in 2018 as a comparison point and to justify
that 3 million m3 was small compared to the overall harvest.
Lines 376, 378
> "It is important to note that prices of wood in neighbouring countries in Central
> Europe were similar (which sug- gests that there was room for a further increase in
> the Polish market)"
If prices in neighbouring countries are similar, there is no need to increase the polish
price. Or are prices higher in neighbouring countries?
Lines 383, 384,
> "domestic and foreign companies showed more interest in the product, which led to
> price reductions [15].
This is counter-intuitive, higher demand should lead to higher prices. Please explain.
> "Research conducted in Japan showed a correlation between lower monthly prices between
> June and August, due to damage caused by pests [107]. Assort- ments with medium
> diameter (S2A) showed greater price stability."
I was really wondering why you mentioned this Japanese research at this point of the
discussion. Then I remembered that you describe S2A as "general-purpose cordwood S2AP
with the permissible level of rot at up to 50%" in the introduction. It's probably worth
mentioning here that S2A is an assortment likely to contain high levels of salvage
loggings after storm or insect damage (if I understand well). And then the comparison
with this Japanese study on pest damage makes sense.
> "Attention should also be paid to the situation to the East of Poland. Among the main
> suppliers of wood residues were countries such as Ukraine, Russia, and Belarus. Due to
> the war situation across Poland's Eastern border, supplies have been significantly
> restricted. The reduction in resources due to the discontinuation of imports from the
> East has also been the cause of the increase in the prices of wood raw materials
> destined for biomass."
Is there a way to control for the effect of supply restrictions due to the war? Please
provide an idea of the polish import volumes from these 3 countries (such as an average
over the 5 years before 2021) and compared to the polish imports from the rest of the
world and to the domestic production in Poland.
Lines 414, 415
> "The verification of time series was used to analyse the general variability of price
> components, depending on the parameter causing fluctuations in the price and in the
> time horizon of the change."
This sentence doesn't make sense, please clarify. Is this verification about the
stationarity test?
Lines 422, 424
> "The local character of the occurrence of random distortions did not have a
> significant impact on the price of these assortments."
Do you have access to more data points that the single quarterly price values you are
showing?
If yes, consider using panel data methods, they have greater power to identify patterns.
Also if you have more data points each quarter, how did you compute the historical
prices? Are these mean or median values? What is the distribution? Can you provide an
idea of the first and the third quartile for example?
Lines 437 438
> "Long-term price fluctuations displayed a minor upward trend, while the predicted
> price of wood was distinguished by a clearer upward tendency
Long-term price fluctuations displayed a minor upward trend, while the predicted
price of wood over last 2 years was distinguished by a clearer upward tendency which
continued over the forecasting period.
Lines 439 to 441
> "In the long term, the growing demand for medium-sized wood will be causing a rise in
> prices as a result of oversupply."
This doesn't make sense. Oversupply usually creates a drop in prices. You can say that
the growing demand can cause a rise in prices. Stop there.
You could update the conclusion to highlight the strengths and weaknesses of your
approach along the following points:
- Prices are the result of complex mechanisms which you illustrated well in your
article. An ARIMA model captures past market behaviours and reproduces them in the
future. Based on a single price series, the model captures how prices tend to progress
through economic cycles and seasonality. This low data requirement is a strength of
the ARIMA approach in a forecasting situation with so many unknowns.
- Relying on a single time series of past prices is also a weakness in that it is not
capable to take into account future changes in market structures due to exogenous
drivers: fuel prices, pulp prices, trade restrictions, natural disturbances etc.
- highlight the rapidly increasing uncertainty of price forecasts through time. You
could end with a recommandation that the forecasting is useful for policy making on
the short term, one or 2 quarters. And that other methods, that take into account
exogenous drivers have to be used beyond that.
Bibliography
Lines 456 and 592, Citation 2 and 62 are identical. Keep only one.
Author Response
Reviewer 2
Thank You very much for your thorough evaluation of our publication. Your comments and corrections are very valuable. They represent a significant improvement in the quality of the publication. We hope that the present explanations will be satisfactory to You.
With best regards
Authors
Comments and suggestions for Authors
The beginning of the article provides a pleasant introduction to the various issues
impacting the polish forest sector and the discussion sections provides a lot of
interesting insights.
My main concern is that the forecasting model is sensitive to the large price increases
in recent months. The article would benefit from a sensitivity analysis on this aspect.
This would require splitting the data into a training and a test set and comparing the
output of your prediction on the real test data, more details further down below. The
small number of observations (18) is also a concern for the validity of the approach. It
would be useful to have time series data that represents several economic cycles.
The authors should make a greater effort at taking the influence of the rest of the
economy into account. In particular, the forecasting model could be improved with the
following information on the past (training) side of the observations:
1. correcting price series for the effect of inflation
Response to the review: At the stage of developing the methodological assumptions, the authors considered including the inflation rate in the source materials used to prepare forecasts. Taking into account the fact that the forecasts generated using the proposed method are to be used for utilitarian purposes, it was concluded that the inflation factor is a component of the analyzed prices. In our research, we adopted the definition that the price is a numerical expression of the value of money that should be spent on the purchase of a given good at a given time. So we're predicting a nominal value that takes inflation into account. It should be emphasized that the article is the beginning of the search for appropriate methods of predicting price changes on the biomass market and the obtained results are to be used in the future as the basis for further research. In the series of subsequent articles, we will also carry out research taking into account the real price. Using the material presented in this article, we will obtain a comparative base as a source material. We hope that the source data and forecast results
controlling for the effect of natural disturbances such as storms and insect damages
followed by salvage logging
In practice, there is the phenomenon of risk, which can be assessed using appropriate methods, and the phenomenon of uncertainty related to exogenous factors, in particular abiotic ones, occurring non-specifically. The use of nominal prices in the time series takes these random factors into account. For example, in Poland in the period under review, the effects of drought and insect outbreaks were felt, which caused an increase in the supply of wood raw material, and thus biomass. This undoubtedly had an impact on the nominal prices that were used as source material. Although these factors have not been identified, it should be considered that they have been included in the price reflecting the current situation on the market. It should be emphasized that the aim of the work was to identify the possibility of using information on the nominal price to predict the nominal price in future periods. The area of ​​research indicated by the reviewer is an extremely interesting thread consisting in the identification of non-market factors in the formation of prices for wood biomass. In subsequent research, we will take this issue into account by identifying and quantifying such variables as an element that allows for even more accurate forecasts. However, a reference point is needed to interpret this type of research. Such a point is undoubtedly the results of the research presented in this article. We would like to thank the reviewer for pointing out the potential possibilities of using the presented research results in further stages of price prediction research. The authors hope that the initiated research, after carrying out the entire research process, will allow to obtain a homogeneous method dedicated to the biomass market. For this to happen, however, the entire research process must be presented, and one of its elements is the proposed article identifying the possibility of using the ARIMA method in the nominal price time series for predicting nominal prices in future periods. From the point of view of utilitarian use of it by the industry, such information is of great importance.
3. looking at price transmission with the global prices of fuel wood and roundwood
(using unit prices of trade from a global trade database)
The forecasting model could also be improved on the future (prediction) side, by feeding
it exogenous variables, such as:
a. forecasts of crude oil or natural gas prices
b. forecasts of pulp prices
We fully agree with the reviewer's opinion and would like to inform you that research on the cross elasticity of demand is currently being conducted, which allows to indicate the strength of the relationship between biomass and other economic goods. We intend to present the results of the mixed demand elasticity of biomass with substitute and complementary goods in order to determine the strength of the relationship between them and the analyzed biomass. After presenting them in the further stages of the work, we will implement the reviewer's postulate. As mentioned earlier, the results obtained in subsequent studies will be compared to those presented in the proposed article.
It would be a lot of work to add all the point above, so I will not insist on it. I
strongly encourage you to preform the sensitivity analysis and the inflation correction.
These methods are within reach and should be added to the article. The other points are
more difficult, but considering the very clear influence of exogenous drivers, the
author should at least describe the limits of their forecasting model.
The topic of forecasting the price of wood raw material in Poland is an evolving one. The choice of ARIMA methodology is covered by numerous literature items, where the method has been used in many other fields, including with a strongly changing external area, e.g., COVID-19 mortality forecasting.
Corrected:
For the purpose of the forecast sensitivity test, actual prices for the third quarter of 2022 were taken into account, as this information was reported by PGL LP. The re-maining periods are not finished, which makes it impossible to provide final data.
Table 5. MAE, MAPE, RMSE results for Q3 2022. Source: Own elaboration based on data obtained from the State Forests IT System (SILP).
Assortments |
M2ZE [pln/m3] |
M2 [pln/m3] |
S2AP [pln/m3] |
S4 [pln/m3] |
2022 IIIQ real |
90 |
31 |
232 |
119 |
2022 IIIQ forecast |
131 |
31,79 |
272,93 |
123 |
MAE |
41 |
0,79 |
40,93 |
4 |
MAPE [%] |
0,455556 |
0,025484 |
0,176422 |
0,033613 |
RMSE |
41 |
0,79 |
40,93 |
4 |
The results of the sensitivity test for the use of the ARIMA model to forecast forest bi-omass prices are presented above. The largest errors are characterized by the M2ZE assortments of about 0.41% of the relative percentage error, while the smallest error was found in the M2 assortment, where MAPE was 0.025%.
I don't think the article can be considered as a proof of the effectiveness of the
forecasting as written at the end of the abstract lines 31 to 33:
> "Based on the research we have conducted, we proved that forecasting the prices of the
> analysed assortments using the ARIMA and SARIMA models is effective."
Considering the high uncertainty, I think the word "proof" is too strong here. A first
attempt of illustrating the fitness of the model would be to perform the sensitivity
analysis requested below.
After analyzing the reviewer's comments, we decided to extend the research with the part related to the verification of the obtained forecasts with the actual level of nominal prices in the period for which the forecast was made. Forecasts prepared for the third quarter of 2022 verified on the basis of the actual price level obtained on the biomass market in this period.
The research provides a baseline forecast of timber prices in Poland. The authors plan to further expand the forecasting model to include external factors, such as natural disasters caused by insects or weather conditions. The hint of the impact of natural gas and oil prices is very useful due to the nature of timber price determination, which includes the cost of harvesting the raw material, conditioned by the price of fuels.
An element of inflationary factor in the price of the raw material is included. The authors used data in which this factor affects the proposed price. The effect of inflation on the price of timber can be inferred from the incurred cost of timber, which has increased under the influence of political and economic factors (e.g., the price of fuel, the cost of machinery, etc.).
Lines 80, 82
> "In recent years, the share of small wood M2 and medium-sized wood S2A assortments
> (25%) in the total amount of wood sold by PGL LP has been on the increase [20].
Could you specify why there was an increase of small an medium size wood assortments in
Poland?
Could there be an influence of the salvage logging related to bark beetle?
You mention salvage logging later lines 365-368:
> "Natural disasters [...] were local and did not impact the supply of wood on a
> national scale, with the exception of spruce stands in the years 2017-2018 (degraded
> by bark beetle)"
The study did not take into account the analysis of the structure of bark beetle damage, only the analysis of the usefulness of the ARIMA model in the forecast of forest biomass prices in Poland. However, this is an interesting point that is worth expanding in the next stages of research.
Lines 85, 86
> Adjusting prices to the level of supply can be considered for optmising the benefits
> of price fluctuations [23,26].
In economic theory, price is the result of a market equilibrium between market
participant. In such models, it is assumed that consumers have different willingness to
pay and producers different willingness to accept. The demand (or supply) quantity at
any given price draws a demand (or supply) curve. The crossing point of the demand and
supply curves represents the equilibrium price. You briefly mention such studies lines
The sales form in Poland has a specific character. It results from the monopoly of the State Forests as the manager of about 80% of the timber resources. The form of sale in Poland has a specific character. It results from the non-commercial monopoly of the State Forests as the manager of about 80% of the timber resources. The price of raw material is shaped by activities aimed at maintaining economic balance between the timber producer and the industry. It also has a vertical character resulting from demand for wood products. The process of obtaining average raw material prices, however, was the result of sales turnovers during the year. This reduces the flexibility of price adjustments while allowing to plan sales and production.
96, 97:
> "most studies being based on models of supply and demand in various geographical
> regions"
Since there is only one large supplier in Poland, the "State Forests - National Forest
Holding" would you describe this situation as a monopoly, or a quasi monopoly? In this
situation price setting is not done through a market equilibrium but through some other
mechanism. Could you please provide more details on the mechanism for adjusting prices
in that context?
Indeed, the specificity of forest management indicates the functioning of a statutory monopoly in its specific form of non-profit monopoly. It should be emphasized that, in accordance with the applicable legislative principles, Art. 50 of the Act on Forests, the main player shaping the market operates on the basis of the principle of self-financing, implementing 5 basic goals, of which 4 goals are ecological goals (with the advantage of costs over revenues) and only one of the goals is the production of wood raw material. The four main objectives are mainly financed from this measure. Summing up, there is no formal pursuit of profit maximization in the operation of PGL LP. Legislative solutions related to the sale of wood, as a result, create the basis for the functioning of free market mechanisms. Of course, there is no question of perfect competition here and we still remain in the monopoly system, bearing in mind that it is a monopoly which is supposed to generate revenues at the level of costs. It should be emphasized that we are not talking about a planned economy here, as evidenced by the fact that PGL LP has been generating significant profits from the sale of wood for many years. This is due to the functioning of the principle of free-market price formation, even on the market where the access of subsequent (new) entities is limited. It should be mentioned here that the main distribution channel for wood raw material is the so-called portal - forestry wood where about 80 (currently 70%) of wood raw material is sold. It should be emphasized that sales are carried out separately by 429 units, and on the supply side (on this closed market) there are about 6,000 units. timber buyers. As a result, there are market activities on this market aimed at determining the final purchase and sale price, which gives grounds for conducting research aimed at predicting the direction and level of changes in the price of wood biomass.
You use a forecasting method based purely on past price data, it is important to make
sure that the price setting mechanism didn't change throughout the period. Do you have
indication that this was the case?
Since 2007, the method of selling wood in Poland has been based on the analyzed principles. We confirm that the pricing mechanism has not changed significantly in the period under review.
Related to this, you wrote about market distortion lines 289, 292:
> "Price fluctuations on the market are distorted by the single dominating entity, which
> is PGLLP (responsible for 80% of managed forests in Poland), while all minor entities
> have to adjust their price strat- egies to the dominating entity."
See the explanation of verse 96, 97:
Lines 78, 80
> "decisions of the Director General of the State Forests [20], which regulate the rules
> for signing contracts with enterprises as part of the so-called "purchase history" and
> free access to purchases at higher price levels"
It seems like wood buyers from the "purchase history" contracts can buy at a
preferential, lower price. Do you distinguish low level from high level purchases in
your analysis? What is the proportion of sales in the two markets ("purchase history"
contracts vs high price level purchases? As prices increased, has the proportion of high
level purchases changed throughout the study period?
Answering the question about contractors' purchase histories, it is not possible to extract the information, as data protection documents do not allow the presentation of such data.The pool of timber intended for customers with history is divided in the proportion of 70 to 30. This means that 70% of the annual pool of timber intended for sale is intended for clients with a history of transactions with PGL LP, while 30% of timber is intended for clients increasing their processing capacity wood as well as for new entrepreneurs. The price is determined based on historical data.
It seems this market has two regimes, a fixed price regime and a free floating price
regime. Could you characterize them a little more?
The fixed price applies to prices based on purchase history (currently 70% of the volume of wood purchased in the previous year) and 30% of the price released by auction.
The figure 1 says "Source: Own elaboration based on data obtained from the State Forests
IT System (SILP)". What level of details do you have in that source? Can you give the
number of observations per assortment?
For each assortment, we have 18 observations from the period 2018-2022, taking into account quarterly prices corresponding to the reporting of financial information in PGL LP
Would it be possible to distinguish and separate these "purchase history" and free floating segment of the markets in your analysis?
Unfortunately, the prices of historical purchases are given together with the prices of the free floating segment. This is due to the averaging of available data and limiting access to the values presented by the forest manager.
You could mention the literature on vertical price transmission (or lack thereof). That
literature is concerned with the study of how price increases in the intermediate or
final product sectors (sawnwood, panel, paper), get transmitted by back to the industry
and then transmitted back to the roundwood prices.
Supplemented by
Raw timber prices are linked to vertical price transmission. However, due to the sales procedures carried out by the forest manager, taking into account periodic sales cycles, their impact is significantly delayed. The impact factors are price changes in global and regional markets [1-4].
Ning Z. and Sun C. 2014. Vertical price transmission in timber and lumber markets. J. For. Econ. 20(1): 17–32. https://doi.org/10.1016/j.jfe.2013.07.002
Klepacka A.M., Siry J.P., and Bettinger P. 2017. Stumpage prices: a review of influential factors. Int. For. Rev. 19(2): 158–169. https://doi.org/10.1505/146554817821255114
da Silva B.K., Schons S.Z., Cubbage F.W., and Parajuli R. 2020. Spatial and cross-product price linkages in the Brazilian pine timber markets. For. Pol. Econ. 17: 102186. https://doi.org/10.1016/j.forpol.2020.102186
Jianbang Gan, Tian, N.; Choi, J.; Pelkki M.H. Synchronized movement between US lumber futures and southern pine sawtimber prices and COVID-19 impacts. Canadian Journal of Forest Research 2022. 2021-0326. https://doi.org/10.1139/cjfr-2021-0326
Line 119
Please explain the meaning of alpha_t, z_t, a_t,
It is a good practice to give an equation number.2
Corrected
random disturbance
random disturbance at the moment t-1
Line 122
> "Is a function related to the non-seasonal autoregressive parameter p."
Add "and where" to introduce the next equation. Suggestion:
Is a function related to the non-seasonal autoregressive parameter p and where
Corrected
Is a function related to the non-seasonal autoregressive parameter p and where
Line 130
Please explain "wraz"
Corrected
It was mistake in polish word
Lines 153, 154
> "The data include quarterly data for the period 2018 - second quarter of 2022, which
> provides exactly 18 observations for the forecast."
18 observations is a very short time series for forecasting. Could you describe how the
small sample size is likely to affect your result?
Is it possible to use pre-2018 data? How far could you go back in time with the data
source used? Please perform the analysis with a longer time series.
The amount of data is conditioned by the fact that it was not until 2021 that the Director General of the State Forests indicated the wood sorting’s appropriate for the forest biomass market under the cascading use of wood raw material. In addition, in 2017 there was a strong hurricane in the western part of Poland, which caused huge losses of full-value timber. In 2018, a large part of the post-hurricane wood was designated as firewood (S4), and the residue was allocated to biomass wood. The supply of forest biomass has definitely increased and has begun to be sold in larger quantities. Data prior to 2018, will disturb the prediction analysis.
Lines 156, 157
> "Figure 1. Quarterly prices for M2 ZE, M2, S2AP, S4 assortments"
Using lines instead of bars for the price chart would make it easier to see price
trends.
Corrected
Lines 160, 162
> "Time series methods ensure good results while requiring modest input data, which
> makes them especially useful in the analysis of forest management issues [57]."
For sure it is easier to use only "modest input data" to forecast future wood prices,
but wood prices are also influenced by the rest of the economy as you wrote lines 54, 55
and in the discussion section. To capture these influences in your model, you could add
exogenous variables. In particular - since you focus on fuel wood - it would be
interesting to look at price forecasts of crude oil or natural gas, as they are
potential substitutes for biomass use.
Please explain if possible how you would add exogenous variables to your model and
explore a little bit the literature linking biomass prices to fossil fuel prices.
Due to the nature of basic research on the subject of forecasting wood prices in Poland, the authors plan to further expand the forecasting model with external factors such as natural disasters caused by insects or weather conditions. A hint about the impact of natural gas and oil prices is very useful due to the nature of determining the price of wood, which includes the costs of obtaining the raw material, conditioned by the price of fuels.
Lines 196, 198
> "Prices of residual wood ranged between 20 and 30 PLN/m3. We can observe a signif-
> icant increase in the price in Q4 of 2021, which continued in Q1 and Q2 of 2022. This
> factor influences the forecasting procedure for the data provided."
As explained in "An introduction to statistical learning: with applications in R"
by G James, D Witten, T Hastie, R Tibshirani - 2013, could you please split your input
data into a training set and a test set? This would enable you to test the validity of
your forecast against real data. Please perform the ARIMA estimations by removing more
and more data from the training set:
1. removing the Q4 of 2021
2. removing also the full data from 2021
3. removing also the data from 2020 and 2021
Then generate the price forecasts for the M2ZE, M2, S2AP and S4 assortments again and
provide these as additional plots with doted lines or some way to distinguish the 3
removing steps above. You may leave these plots as annexes to illustrate the effect of
recent price hikes on the increased prediction.
This sensitivity analysis is also related to my previous comment on the length of the
time series. If you can get more data before 2018, then this would be better. It is also
important to consider the number of fixed contracts "" versus free floating contract
throughout the period.
The reviewer's comments are valuable, and we will certainly apply the indicated testing methodology for future research on forecasting the price of sawntimber and valuable timber.
Lines 210, 2013
> "The values of the Point Forecast for residual wood are on an upward trend. It is a
> significant hike in the price rise compared to 2021. Comparing the prices in the first
> two quarters of 2022, which increased by an extremely large amount, gives rise to the
> antici- pation of a further increase in the following years."
How likely it is that the price rise will continue? Could it be that prices revert to
some lower levels. You may also add that uncertainty increases with time i.e. the
forecast for Q1 of 2023 is much more uncertain than the one for Q3 of 2022. By the way
Q3 2022 is real data now, how easy is it for you to update this data point?
It would also be important to compensate for inflation. Please use a measure of national
inflation to correct the price series for the effect of inflation. If the effect is
small, you may remove it in the final analysis, but it is important to take this into
account.
Corrected and the explanation in point 1.
Figures 2,4,6,8
You could simplify the x axis by removing the repeated years and the "kw" symbol (what
does kw mean?).
Corrected for Quarter
Figures 3,5,7,9
Displaying all years on the x axis would make it easier to read. In general the x axis
of all graph could be harmonized.
In this Program is impossible to do that.
Lines 269
> "the SARIMA model, expanded by the seasonality factor, was applied"
Describe how the SARIMA model differs from ARIMA in terms of additional variables in the
equations lines 118 to 129.
Corrected
It was correct in text:
SARIMA(p, d, q)(P, D, Q) model: ARIMA model that takes into account the seasonality component (P – order of seasonal lags of the AR type, Q – order of seasonal lags of the MA type, D – differentiation of the seasonal component).
Lines 276, 279
> "The global climate change that is being observed, the adjustment of regulations, and
> the increased logging caused by biotic and abiotic factors, are the main factors
> influencing fluctuations in wood trade in the last decade."
Modelling the increase of these external influences would require introducing exogenous
variables wouldn't it?
Of course, we agree with this statement of taking into account exogenous factors. This is in the plans for further stages of analysis.
Lines 284, 286
> "Due to these changes, there is currently strong competition in the firewood and
> industrial wood markets (pulp and paper industry), which leads to structural changes
> in the supply of wood."
Since pulp markets are globalized - and considering there is some level of transmission
from pulp prices to roundwood prices - pulp forecasts are another candidate for an
exogenous variable that could drive your price forecast.
Thank you for a very good tip. It will be taken into account in the next stages of research.
Note very right. The impact of vertical price transmission and horizontal has an impact on the response of Polish timber price markets.
The delay in the price response to the behavior of the markets flattens the effect of the price change.
Lines 298 to 302
> "monitoring of the development of the market of wood consumers in the country, as well
> as on the European and global markets, on the basis of the assessment of wood trade
> levels; making trade agreements; conducting assessments at the appropriate frequency,
> and reacting to these changes."
Consider whether the notions of vertical price transmission and horizontal price
transmissions would improve the way you describe the reactions of the polish wood price
markets.
Raw timber prices are linked to vertical price transmission. However, due to the forest manager's sales procedures with periodic sales cycles, their impact is significantly delayed. These factors in the modeling for the studied LP management and are consistent with the change in prices in world markets and neighboring countries. There is impact between the frequency of transactions and the change in timber prices.
Lines 313, 315,
> "Cyclic price patterns in the Polish market were largely connected with the condition
> of the national, European, and global economy, and constituted the main factor
> responsible for general wood price fluctuations."
If this is the case, then you should try hard to get a longer time series. What is the
length of the economic cycle in the literature you cite? It would be useful to have
time series data that represents several economic cycles, at least two or more cycles.
Longer term data would train your ARIMA model to be better at predicting the likely
decrease of prices, once the economy moves into a new phase of the economic cycle.
In the case of forecasting prices for other roundwood grades, the reviewer's statement is implemented. The data collected includes sales cycles during the year ( by quarter). The economic (breeding) cycle in forestry covers a period of about 60-100 years, from the time the tree is planted to the time the tree is felled and the sorties are harvested. An additional complication is the periodic separation of new assortments. The surveyed sorties were determined in the surveyed years (the exception being S2AP biomass in 2021, for example) by the Director General of the State Forests. They differ from price reports and assortments requirements in previous years.
Lines 317, 318
> "to the increase in the supply of raw materials in North America"
I don't understand, is this an increase of wood exports from Poland to North America, or
an increase of imports from North America into Poland? Please clarify.
Thank you for the right comment. The increase in exports was for lumber.
Corected:
During the pandemic and the increase in demand for lumber in North America, the major lumber processors in Europe increased their lumber exports. European prices increased significantly due to the shortage of sawn materials, which translated into increased demand for roundwood [1, ].
Jianbang Gan, Tian, N.; Choi, J.; Pelkki M.H. Synchronized movement between US lumber futures and southern pine sawtimber prices and COVID-19 impacts. Canadian Journal of Forest Research 2022. 2021-0326. https://doi.org/10.1139/cjfr-2021-0326
Riddle, A. 2021. COVID-19 and the US timber industry (updated 29 July 2021). Congressional Research Service, Washington, D.C.
Lines 324, 325
> "A cycle was identified for the analysed period of 3 to about 4 years."
How was the cycle length identified? The price series are just 4 years long from 2018 to
2022. This calls again for a time series that is several times longer than the economic cycle.
It was mistake in methods. Corrected in text
Lines 329, 303
> "due to the growing demand from industry, especially construction."
You could add that construction and renovation increases are related to government stimulus packages following the COVID pandemic as well as to the low bank interest rates at the time.
Thank you very much for this prompt.
Corrected:
The growth in construction and renovation was linked to the government's stimulus packages after the COVID pandemic, as well as low bank interest rates at the time
Lines 331, 334,
> "as a downward trend in wood sales has been historically observed in the period of
> slower economic growth too. Usually, the drop in supply would coincide with upward
> price cycles."
Not very clear, is this a drop in supply due to a supply constraints? Does demand remain
constant in that context? Try to improve.
Corrected:
The decline in demand for wood products was seen during periods of slower economic growth. This led to a decrease in timber harvesting. This consequently caused fluctuations in supply and increases in demand, which coincided with gradual cycles of price changes.
Lines337,338
> "Due to the volatility in global markets and the significant drop in demand in 2019,
> the State Forests managed to fulfil its sales plan, albeit at slightly reduced
> prices."
A verb is missing from this sentence. Or maybe you should start the sentence with
"despite" instead of "Due to".
Corrected
Lines 342
> "Industrial wood (M2, M2ZE, S2AP, and S4)"
The definitions given lines 147, 150 don't correspond to the FAO definition of
industrial roundwood. M2, M2ZE, S2AP, and S4 are mostly fuel wood products aren't they?
See the FAOSTAT Forest Product Production Statistics – Data Structure
> "residual wood M2ZE, firewood slash M2, medium-sized firewood S4 with the permissible
> level of soft rot set at 50%, minimum top diameter of 5 cm without bark, and
> general-purpose cord- wood S2AP"
It seems these assortments correspond to fuel wood types. Please clarify or explain why
you put them under industrial roundwood.
Taking into account the biomass assortments presented in the publication, the quality and dimensional classification applicable in Poland was followed, which in 2021 were recognized as energy wood by the Director General of the State Forests.
Lines 371, 372
> "resulted in an additional supply of about 3 million m3 of wood, also in 2018"
Please add the overall Polish wood supply in 2018 as a comparison point and to justify
that 3 million m3 was small compared to the overall harvest.
Corrected
“…resulted in an additional supply of about 3 million m3 of wood, also in 2018 (the annual harvest of wood in Poland is about 39 million m3) [110].”
Lines 376, 378
> "It is important to note that prices of wood in neighbouring countries in Central
> Europe were similar (which sug- gests that there was room for a further increase in
> the Polish market)"
If prices in neighbouring countries are similar, there is no need to increase the polish
price. Or are prices higher in neighbouring countries?
Corrected
As stated earlier, timber prices in Poland are set in rounds of sales (contract). Other specific auction sytems include periodic information about the inclusion of harvesting, skidding and administration costs in the minimum price. The setting of maximum prices in previous years was often intended to ensure the stability of the timber industry in Poland.
Lines 383, 384,
> "domestic and foreign companies showed more interest in the product, which led to
> price reductions [15].
This is counter-intuitive, higher demand should lead to higher prices. Please explain.
Note and corrected
When responding to the issue of increased demand and reduced prices, one should remember about the closed nature of the timber market in Poland. Prices have been lowered for Polish entrepreneurs who are the main recipients of wood, where such large fluctuations that have taken place recently have affected the overall price of wood raw material. For companies in the wood industry, these prices were too high, which resulted in the lack of profitability of doing business. Such decisions could have huge effects in the Polish economy, where the wood industry plays an important role.
> "Research conducted in Japan showed a correlation between lower monthly prices between
> June and August, due to damage caused by pests [107]. Assort- ments with medium
> diameter (S2A) showed greater price stability."
I was really wondering why you mentioned this Japanese research at this point of the
discussion. Then I remembered that you describe S2A as "general-purpose cordwood S2AP
with the permissible level of rot at up to 50%" in the introduction. It's probably worth
mentioning here that S2A is an assortment likely to contain high levels of salvage
loggings after storm or insect damage (if I understand well). And then the comparison
with this Japanese study on pest damage makes sense.
The remark is correct, but it is the period of increased insect activity that promotes damage to stands. This requires accelerated timber harvesting. The example of Japan is a reference of the increase in intensity of attack.
The indication S2A applies to wood in Poland, which as already damaged is not subject to such a strong decline in value as other roundwood sorts.
Similar activities can be seen in Europe, however, due to the assumptions of sustainable development, European procedures are longer and not so strongly exposed. However, attention is included and the impact of degradation in European countries is indicated.
- Fernández-Fernández, M.; Naves, P.; Musolin, D.L.; Selikhovkin, A.V.; Cleary, M.; Chira, D.; Paraschiv, M.; Gordon, T.; Solla, A.; Papazova-Anakieva, I.; Drenkhan, T.; Georgieva, M.; Altunisik, A.; Morales-Rodríguez, C.; Tabaković-Tošić, M.; Avtzis, D.N.; Georgiev, G.; Doychev, D.D.; Nacheski, S.; Trestic, T.; Elvira-Recuenco, M.; Diez, J.J.; Witzell, J. Pine Pitch Canker and Insects: Regional Risks, Environmental Regulation, and Practical Management Options. Forests2019, 10, 649. https://doi.org/10.3390/f10080649
> "Attention should also be paid to the situation to the East of Poland. Among the main
> suppliers of wood residues were countries such as Ukraine, Russia, and Belarus. Due to
> the war situation across Poland's Eastern border, supplies have been significantly
> restricted. The reduction in resources due to the discontinuation of imports from the
> East has also been the cause of the increase in the prices of wood raw materials
> destined for biomass."
Is there a way to control for the effect of supply restrictions due to the war? Please
provide an idea of the polish import volumes from these 3 countries (such as an average
over the 5 years before 2021) and compared to the polish imports from the rest of the
world and to the domestic production in Poland.
Corrected:
“The import of pellets from Ukraine is of great importance for Poland. Our country was the largest importer of this product - in 2021 we imported a total of 121,000 tonnes from Ukraine, which accounted for nearly 30% of all Ukrainian exports (412,000 tonnes). Considering that the volume of pellet production oscillates around 1 million tonnes, this is quite a serious loss.”[1]
- Pellet market in Europe after pressure of war in Ukraine. https://www.drewno.pl/artykuly/12552,rynek-peletu-w-europie-po-presja-wojny-na-ukrainie.html (Accesed on 29 Nowember 2022)
Lines 414, 415
> "The verification of time series was used to analyse the general variability of price components, depending on the parameter causing fluctuations in the price and in the time horizon of the change."
This sentence doesn't make sense, please clarify. Is this verification about the stationarity test?
Corrected:
"The verifiable temporal distribution was used to analyze the overall price volatility of wood raw material."
Lines 422, 424
> "The local character of the occurrence of random distortions did not have a
> significant impact on the price of these assortments."
Do you have access to more data points that the single quarterly price values you are
showing?
If yes, consider using panel data methods, they have greater power to identify patterns.
Also if you have more data points each quarter, how did you compute the historical
prices? Are these mean or median values? What is the distribution? Can you provide an
idea of the first and the third quartile for example?
At the moment we have no information on the dispersal of local bark beetle gradations. After a general understanding of the size of the areas affected by the insect, the authors found no impact on the overall price of the presented assortments.
Historical prices were obtained from PGL LP.
Lines 437 438
> "Long-term price fluctuations displayed a minor upward trend, while the predicted
> price of wood was distinguished by a clearer upward tendency
Long-term price fluctuations displayed a minor upward trend, while the predicted
price of wood over last 2 years was distinguished by a clearer upward tendency which
continued over the forecasting period.
Corrected
Lines 439 to 441
> "In the long term, the growing demand for medium-sized wood will be causing a rise in
> prices as a result of oversupply."
This doesn't make sense. Oversupply usually creates a drop in prices. You can say that
the growing demand can cause a rise in prices. Stop there.
You could update the conclusion to highlight the strengths and weaknesses of your
approach along the following points:
- Prices are the result of complex mechanisms which you illustrated well in your
article. An ARIMA model captures past market behaviours and reproduces them in the
future. Based on a single price series, the model captures how prices tend to progress
through economic cycles and seasonality. This low data requirement is a strength of
the ARIMA approach in a forecasting situation with so many unknowns.
Corrected of conclusion
- Relying on a single time series of past prices is also a weakness in that it is not
capable to take into account future changes in market structures due to exogenous
drivers: fuel prices, pulp prices, trade restrictions, natural disturbances etc.
- highlight the rapidly increasing uncertainty of price forecasts through time. You
could end with a recommandation that the forecasting is useful for policy making on
the short term, one or 2 quarters. And that other methods, that take into account
exogenous drivers have to be used beyond that.
>
Good point. The transcription error has been corrected and removed in the task:
“In the long term, growing demand for medium-sized timber will drive up prices”
conclusions corrected:
Price forecasting is the result of complex mechanisms. The ARIMA model captures past market behavior using a single data series. This low data requirement is a strength of the ARIMA approach in a forecasting situation with many unknowns.
- The weakness of the model is its reliance on a single time series of prices from forecasting. It does not allow for the inclusion of future changes caused by exogenous factors: transportation prices, trade restrictions, natural disasters, etc.
- Forecasts are useful for creating a short-term forecast, one or two quarters. for use beyond this period, other methods are needed that take into account exogenous factors
Bibliography
Lines 456 and 592, Citation 2 and 62 are identical. Keep only one.
Corrected
We thank the Reviewer for important comments that enhance the work.
The suggestions are very pertinent and will help in the development of future articles.

Reviewer 3 Report
Thank you, editor and author(s), for providing an opportunity to read this manuscript entitled “Biomass price prediction on the example of Poland.” The paper is well-written and interesting for policymakers, forest managers, and forest owners.
Comments:
- The forecast has been done with the 18 observations. Is using the ARIMA model with fewer than 30 observations statistically valid? Please provide justification with the support of the literature.
- Which ARIMA model was used in this study? For instance, Box-Jenkins model. Also, briefly describe the theoretical steps of that model.
- Please define M2ZE, firewood slash M2, and medium-sized firewood mathematically. It means what is the length and diameter of those classes of woods, and what is the significance of those woods in the market?
- Lines 289-301: Inflation plays a crucial role in the price of wood products. So, could you please discuss how inflation affects the future price of wood in Poland?
- Please add in the conclusion section: What would be the implication of the increasing trend of the wood price? Is that going to increase the profitability to forest owners? Will that affect the future wood supply (or availability) in the market in the long and short run?
Specific comments:
- Line 144-146: Citation needed
- Line 182: Scientific name should be italicized.
- Line 290: Please spell PGLLP.
Author Response
Reviewer 3
Thank You very much for your thorough evaluation of our publication. Your comments and corrections are very valuable. They represent a significant improvement in the quality of the publication. We hope that the present explanations will be satisfactory to You.
With best regards
Authors
Comments and suggestions for Authors
Thank you, editor and author(s), for providing an opportunity to read this manuscript entitled “Biomass price prediction on the example of Poland.” The paper is well-written and interesting for policymakers, forest managers, and forest owners.
Comments:
- The forecast has been done with the 18 observations. Is using the ARIMA model with fewer than 30 observations statistically valid? Please provide justification with the support of the literature.
Considering the assortments of biomass presented in the publication, they were guided by the current Polish qualitative and dimensional classification, which in 2021 were recognized by the Director General of the State Forests as energy wood.
Price statements are the result of obtainable data from PGLLP. Previously, some groups of raw material were not separated from the total volume of harvesting in Poland
- Which ARIMA model was used in this study? For instance, Box-Jenkins model. Also, briefly describe the theoretical steps of that model.
We also include a description of the stages of the research being conducted. The inclusion of the data in the article as an appendix is due to the large size of the information.
Corrected
SARIMA(p, d, q)(P, D, Q) model: ARIMA model that takes into account the seasonality component (P – order of seasonal lags of the AR type, Q – order of seasonal lags of the MA type, D – differentiation of the seasonal component).
- Please define M2ZE, firewood slash M2, and medium-sized firewood mathematically. It means what is the length and diameter of those classes of woods, and what is the significance of those woods in the market?
Taking into account the biomass assortments presented in the publication, the quality and dimensional classification applicable in Poland was followed, which in 2021 were recognized as energy wood by the Director General of the State Forests.
M2ZE -branch and chipped wood destined for woodchips for energy purposes with a minimum diameter of at least 5 cm without bark (7 cm in bark), the length or quality of which does not allow industrial use
M2 small-sized firewood up to 7 cm in diameter with a length of 0.5 m to 6.0 m
S4 medium-sized firewood up to 24 cm with a length of 0.5 m to 6.0 m
- Lines 289-301: Inflation plays a crucial role in the price of wood products. So, could you please discuss how inflation affects the future price of wood in Poland?
Response to the review: At the stage of developing the methodological assumptions, the authors considered including the inflation rate in the source materials used to prepare forecasts. Taking into account the fact that the forecasts generated using the proposed method are to be used for utilitarian purposes, it was concluded that the inflation factor is a component of the analyzed prices. In our research, we adopted the definition that the price is a numerical expression of the value of money that should be spent on the purchase of a given good at a given time. So we're predicting a nominal value that takes inflation into account. It should be emphasized that the article is the beginning of the search for appropriate methods of predicting price changes on the biomass market and the obtained results are to be used in the future as the basis for further research. In the series of subsequent articles, we will also carry out research taking into account the real price. Using the material presented in this article, we will obtain a comparative base as a source material. We hope that the source data and forecast results presented in this form will also serve other researchers as a reference point for further analyses. Summing up, we would like to thank the reviewer for drawing attention to the inflation factor, which will be taken into account when preparing source materials for subsequent studies related to the analysis of time series.
- Please add in the conclusion section: What would be the implication of the increasing trend of the wood price? Is that going to increase the profitability to forest owners? Will that affect the future wood supply (or availability) in the market in the long and short run?
Thank you for that opinion. However, this topic has not been the subject of research on the use of the ARIMA methodology to predict timber prices. This hint will be included in the publication cycle on forest biomass in Poland
Specific comments:
- Line 144-146: Citation needed Corrected
- Line 182: Scientific name should be italicized.Corrected
- Line 290: Please spell PGLLP. Corrected
We thank the Reviewer for important comments that enhance the work.
The suggestions are very pertinent and will help in the development of future articles.

Reviewer 4 Report
- Use scatter with line graph to show the price changes.
- Did the price was deflated using consumer price index? I suggest explaining it in the method section.
- The details of stationary test and ACF and PACF graphs can be included in the paper to provide more information.
- The quality of forecasted model by ARIMA is not mentioned in the text. In the Did you have the valuses of root mean square, mean absolute error, mean absolute percentage error and the Theil Inequality Coefficient?
- Language editing is required.
- The quality of forecasted model by ARIMA is not mentioned in the text. In the Did you have the valuses of root mean square, mean absolute error, mean absolute percentage error and the Theil Inequality Coefficient?
- Language editing is required.
Author Response
Reviewer 4
Thank You very much for your thorough evaluation of our publication. Your comments and corrections are very valuable. They represent a significant improvement in the quality of the publication. We hope that the present explanations will be satisfactory to You.
With best regards
Authors
Comments and suggestions for Authors
- Use scatter with line graph to show the price changes.
Corrected
Did the price was deflated using consumer price index? I suggest explaining it in the method section.
Response to the review: At the stage of developing the methodological assumptions, the authors considered including the inflation rate in the source materials used to prepare forecasts. Taking into account the fact that the forecasts generated using the proposed method are to be used for utilitarian purposes, it was concluded that the inflation factor is a component of the analyzed prices. In our research, we adopted the definition that the price is a numerical expression of the value of money that should be spent on the purchase of a given good at a given time. So we're predicting a nominal value that takes inflation into account. It should be emphasized that the article is the beginning of the search for appropriate methods of predicting price changes on the biomass market and the obtained results are to be used in the future as the basis for further research. In the series of subsequent articles, we will also carry out research taking into account the real price. Using the material presented in this article, we will obtain a comparative base as a source material. We hope that the source data and forecast results presented in this form will also serve other researchers as a reference point for further analyses. Summing up, we would like to thank the reviewer for drawing attention to the inflation factor, which will be taken into account when preparing source materials for subsequent studies related to the analysis of time series.
The details of stationary test and ACF and PACF graphs can be included in the paper to provide more information.
The ACF and PACF stationarity test was done to each sorting item to be able to identify specific variants of the ARIMA methodology, e.g. ARIMA (0,2,0). Unfortunately, the too large size of the graphs would technically prevent transparency of the publication. Relative and absolute error values are not calculated due to the lack of up-to-date information on average prices of the analyzed sorties. This is due to the fact that the data were divided into quarterly time series corresponding to the reporting of financial information by PGL LP.
The quality of forecasted model by ARIMA is not mentioned in the text. In the Did you have the valuses of root mean square, mean absolute error, mean absolute percentage error and the Theil Inequality Coefficient?
After analyzing the reviewer's comments, we decided to extend the research with the part related to the verification of the obtained forecasts with the actual level of nominal prices in the period for which the forecast was made. Forecasts prepared for the third quarter of 2022 verified on the basis of the actual price level obtained on the biomass market in this period.
The research provides a baseline forecast of timber prices in Poland. The authors plan to further expand the forecasting model to include external factors, such as natural disasters caused by insects or weather conditions. The hint of the impact of natural gas and oil prices is very useful due to the nature of timber price determination, which includes the cost of harvesting the raw material, conditioned by the price of fuels.
Language editing is required.
Corrected
We thank the Reviewer for important comments that enhance the work.
The suggestions are very pertinent and will help in the development of future articles.

Round 2
Reviewer 2 Report
The paper has improved compared to the previous version. The sensitivity analysis is too
shallow and could be expanded to 3 or 5 data points.
Lines 198,199
> The sensitivity analysis was performed in comparison with the third quarter of 2022,
> as this period can be considered closed and PGL LP reported the data.
Good that you conducted a sensitivity analysis on the most recent data point. I am a bit
disappointed by the fact that you perform the sensitivity on one point only. It would
have been stronger if you had removed at least the last 5 points from the training set
and used those 5 points as a test set.
Lines 155, 161 and 166
> MAE ... MAPE .... RMSE
Great that you provide a mathematical definition in equations 6,7,8. Please give also
the meaning of the MAE, MAPE, RMSE acronyms in the text. For example I guess that RMSE
means "Root Mean Squared Error", what about the other 2?
Line 325 Table 5
> MAPE [%] 0,455556 0,025484 0,176422 0,033613
This is not the result of formula 7. The values are not in percentage but in fraction.
Either remove the `%` sign or multiply by 100 to obtain percentages.
> The largest errors are characterized by the M2ZE assort- ments of about 0.41% of the
> relative percentage error, while the smallest error was found in the M2 assortment,
> where MAPE was 0.025%.
Computing MAE / real price = 41 / 90 = 0.455556
Thus, the error is 45% and not 0.45%.
Other values should also be corrected, it is not 0.025% but 2.5%.
Figures 1,2,4,6,8. My earlier comment concerning the X axis was not fully taken into
account. There are still repeated years. It is just an advice that would enhance the
readability of your plots. I guess those figures are made with Excels, while the other
figures (3,5,7,9) are made with R and I understand that it may not be that easy to
customize the X axis in Excel. Maybe there is a way to specify that the X axis is a time
series variable?
Lines 376-379
> "The years 2018-2022 were a period of dynamic economic change caused by the expansion
> in the economic development of the wood industry (in 2020), which was due to the
> increase in the supply of raw materials in North America at the time of a declining
> long-term trend"
This sentence is strange, check the logic, remove it or break it in 2 sentences.
Lines 379-382
> "During the pandemic and the increase in demand for lumber in North America, the major
> lumber processors in Europe increased their lumber exports."
According to UN Comtrade data, the 10 largest export destinations for Poland over the
last 5 years were all intra EU countries:
reporter partner flow net_weight trade_value
Poland World export 7.284500e+09 4.330319e+09
Poland Germany export 2.818798e+09 1.381841e+09
Poland Czechia export 5.185681e+08 1.896197e+08
Poland Italy export 4.944938e+08 2.400954e+08
Poland Denmark export 3.318245e+08 1.511777e+08
Poland France export 3.287385e+08 3.108926e+08
Poland Slovakia export 3.225158e+08 1.231330e+08
Poland Sweden export 3.106429e+08 1.995911e+08
Poland Lithuania export 3.058335e+08 1.440347e+08
Poland United Kingdom export 2.327580e+08 3.885397e+08
Poland Netherlands export 2.073795e+08 1.531868e+08
Poland Romania export 1.497591e+08 9.219125e+07
It is OK to cite American literature as a reference for increased wood consumption, but
the data seems to indicate that increased consumption inside the EU is the major factor
influencing Polish exports. Above table doesn't take into account re-exports and
concerns only products under HS code 44..
Lines 454,455
> "The setting of the maximum price in previous years was intended to keep the timber
> industry in Poland stable"
Specify again who sets the price. The reader knows that it is the polish state forest
who sets the price so it is just easier to write that the polish state forest sets the
price in order to have a stable timber industry.
Lines 532 to 536. Great that you put the strengths and weaknesses of the ARIMA approach.
Looking forward to your future work plans including exogenous drivers.
I wish you all the best for the publication step.
Author Response
Reviewer 2
Thank you very much for such a thorough review.
It gave the article a high quality and it is a credit to the Reviewer.
We apologize that if we could not meet all expectations.
The comments are valuable and will be used in planning future studies and publications.
Comments to authors
The paper has improved compared to the previous version. The sensitivity analysis is too shallow and could be expanded to 3 or 5 data points.
Lines 198,199
> The sensitivity analysis was performed in comparison with the third quarter of 2022, as this period can be considered closed and PGL LP reported the data.
Good that you conducted a sensitivity analysis on the most recent data point. I am a bit disappointed by the fact that you perform the sensitivity on one point only. It would have been stronger if you had removed at least the last 5 points from the training set and used those 5 points as a test set.
We agree with the reviewer that a larger sensitivity test would have been more satisfactory. Reports are available with a significant delay. However, the study created a forecast for Q3 2022, Q4 2022 and Q1 2023. A larger sensitivity test involves conducting a historical forecast, which would change the entire study. We thank you for your valuable comment and hint, which will be used in future analyses. We count on the understanding of a reviewer experienced in forecasting studies. The guidance will be very useful in determining the data.
Lines 155, 161 and 166
> MAE ... MAPE .... RMSE
Great that you provide a mathematical definition in equations 6,7,8. Please give also the meaning of the MAE, MAPE, RMSE acronyms in the text. For example I guess that RMSE means "Root Mean Squared Error", what about the other 2?
Corrected
Thank you for pointing out the missing explanations. They have been completed in the methodology section.
MAE [Mean Absolute Error]:
[6]
x- real value of price
x0- forecast value of prie
MAPE [Mean Absolute Percentage Error]:
[7]
RMSE [Root Mean Squared Error]:
[8]
N- number of observation
Line 325 Table 5
> MAPE [%] 0,455556 0,025484 0,176422 0,033613
This is not the result of formula 7. The values are not in percentage but in fraction.
Either remove the `%` sign or multiply by 100 to obtain percentages.
> The largest errors are characterized by the M2ZE assort- ments of about 0.41% of the relative percentage error, while the smallest error was found in the M2 assortment, where MAPE was 0.025%.
Computing MAE / real price = 41 / 90 = 0.455556
Thus, the error is 45% and not 0.45%.
Other values should also be corrected, it is not 0.025% but 2.5%.
Thank you for pointing out this part of the results. There was an error in the formula in excel multiplying by 100%. The result was not calculated correctly. MAPE has been corrected in the table in the results section. Many thanks to the reviewer for such a thorough analysis of the results of the study.
The results have been corrected according to the correct scale
Assortments |
M2ZE [pln/m3] |
M2 [pln/m3] |
S2AP [pln/m3] |
S4 [pln/m3] |
2022 IIIQ real |
90 |
31 |
232 |
119 |
2022 IIIQ forecast |
131 |
31,79 |
272,93 |
123 |
MAE |
41 |
0,79 |
40,93 |
4 |
MAPE [%] |
45,56% |
2,55% |
17,64% |
3,36% |
RMSE |
41 |
0,79 |
40,93 |
4 |
The results of the sensitivity test for the use of the ARIMA model to forecast forest biomass prices are presented above. The largest errors are characterized by the M2ZE assortments of about 45,56 % of the percentage error, while the smallest error was found in the M2 assortment, where MAPE was 2,55%.
Figures 1,2,4,6,8. My earlier comment concerning the X axis was not fully taken into account. There are still repeated years. It is just an advice that would enhance the readability of your plots. I guess those figures are made with Excels, while the other figures (3,5,7,9) are made with R and I understand that it may not be that easy to customize the X axis in Excel. Maybe there is a way to specify that the X axis is a time series variable?
In the figures that were created in Excel, the change has been taken into account and indeed the results are clearer. Thank you for the very pertinent advice. In the case of figures created by the R program, this is not easy to do. We will certainly include this division of the X-axis at the beginning in our next work and calculations. It will make it easier for readers to evaluate the results for which we thank the Reviewer.
Lines 376-379
> "The years 2018-2022 were a period of dynamic economic change caused by the expansion in the economic development of the wood industry (in 2020), which was due to the increase in the supply of raw materials in North America at the time of a declining long-term trend"
This sentence is strange, check the logic, remove it or break it in 2 sentences.
Corrected
“The years 2018-2022 were a period of dynamic economic changes caused by an expansion in the economic development of the timber industry (in 2020). In addition to the increased demand for raw material from European Union countries, the increase in the supply of raw material in North America was significant."
Lines 379-382
> "During the pandemic and the increase in demand for lumber in North America, the major lumber processors in Europe increased their lumber exports."
According to UN Comtrade data, the 10 largest export destinations for Poland over the last 5 years were all intra EU countries:
reporter partner flow net_weight trade_value
Poland World export 7.284500e+09 4.330319e+09
Poland Germany export 2.818798e+09 1.381841e+09
Poland Czechia export 5.185681e+08 1.896197e+08
Poland Italy export 4.944938e+08 2.400954e+08
Poland Denmark export 3.318245e+08 1.511777e+08
Poland France export 3.287385e+08 3.108926e+08
Poland Slovakia export 3.225158e+08 1.231330e+08
Poland Sweden export 3.106429e+08 1.995911e+08
Poland Lithuania export 3.058335e+08 1.440347e+08
Poland United Kingdom export 2.327580e+08 3.885397e+08
Poland Netherlands export 2.073795e+08 1.531868e+08
Poland Romania export 1.497591e+08 9.219125e+07
It is OK to cite American literature as a reference for increased wood consumption, but
the data seems to indicate that increased consumption inside the EU is the major factor
influencing Polish exports. Above table doesn't take into account re-exports and
concerns only products under HS code 44..
Corrected
Thank you for the right comment corrected regarding line 376-379
Lines 454,455
> "The setting of the maximum price in previous years was intended to keep the timber industry in Poland stable"
Specify again who sets the price. The reader knows that it is the polish state forest who sets the price so it is just easier to write that the polish state forest sets the price in order to have a stable timber industry.
Corrected
“The setting of the maximum price by the State Forests in previous years was intended to keep the timber industry in Poland stable”
Lines 532 to 536. Great that you put the strengths and weaknesses of the ARIMA approach. Looking forward to your future work plans including exogenous drivers.
I wish you all the best for the publication step.
Thanks again for all your comments and contributions to the quality of the publication
Authors

Reviewer 4 Report
After reading the revised manuscript, I noticed that the authorsrevised it based on my comments. Hence, I recommend the publication ofthis manuscript.".
Author Response
Dear Reviewer 4
Comments and Suggestions for Authors
After reading the revised manuscript, I noticed that the authors
revised it based on my comments. Hence, I recommend the publication of
this manuscript.".
Thank you for all your comments, which helped to significantly improve the quality of the publication.
We hope to publish further studies. The suggestions will allow to significantly improve both the methodology and the description of the research results.
Best regards
Authors